# Wisdom of the Crowd Voting: Truthful Aggregation of Voter Information and Preferences

**Grant Schoenebeck**
School of Information
University of Michigan
schoeneb@umich.edu

**Biaoshuai Tao**[*]
John Hopcroft Center for Computer Science
Shanghai Jiao Tong University
bstao@sjtu.edu.cn

## Abstract

We consider two-alternative elections where voters' preferences depend on a state variable that is not directly observable. Each voter receives a private signal that is correlated to the state variable. Voters may be "contingent" with different preferences in different states; or predetermined with the same preference in every state. In this setting, even if every voter is a contingent voter, agents voting according to their private information need not result in the adoption of the universally preferred alternative, because the signals can be systematically biased.

We present an easy-to-deploy mechanism that elicits and aggregates the private signals from the voters, and outputs the alternative that is favored by the majority. In particular, voters truthfully reporting their signals forms a strong Bayes Nash equilibrium (where no coalition of voters can deviate and receive a better outcome).

## 1 Introduction

Social choice theory studies how to aggregate participants' heterogeneous opinions/preferences and output a collective decision from a set of alternatives. Typically, though not always, it is assumed that each participant has a clear preference over the alternatives, e.g., a preference order over all the alternatives, a valuation for each alternative, etc. However, even with only two alternatives, this is not the typical case for *all* participants. In addition to the participants who have clear, *predetermined* preferences for one alternative over the other, typically there are also *contingent* participants who only have partial information on which alternative is "preferable for them" and yet would like to select the alternative that is "preferable for them."

A standard example would be an election with two candidates $a$ and $b$ coming from political parties $A$ and $B$ respectively. Voters are normally partitioned into three types: some partisans for Party $A$ prefer candidate $a$ based on his support for the platform of party $A$; other partisans for Party $B$ prefer candidate $b$ based on her support for the platform of party $B$; finally, there are swing voters who are largely indifferent between the parties' platforms and would like to elect whichever candidate can make more progress on non-partisan issues. However, swing voters do not have perfect information about which candidate is better suited for addressing the non-partisan needs of the community. Instead, each voter has a hunch of which candidate will perform better on the non-partisan problems facing the community based on both public information and their private experiences and beliefs.

Additional examples where participants have preferences, but may or may not know what is "preferable for them", abound. In votes for corporate strategies, for hiring decisions, and for policy decisions typically some participants would like to select according to some truth they are collectively trying to

---

A full version of this paper is available at https://arxiv.org/abs/2108.03749.
This material is based upon work supported by the National Science Foundation under Grant No. 2007256.
[*]corresponding author

discern (e.g., impact on future profits, suitability for the position, efficacy of policy, etc.) while others may have predetermined preferences, for example, because of the way they are uniquely affected (e.g., prominence of their position in future corporate strategy, vision/skills of the job candidate, who in particular the policy benefits/harms).

## 1.1 Informal Setting

In this paper, we consider a two-alternative social choice setting where voters' preferences may depend on a state variable. In the above example, the state is which candidate will make more progress on non-partisan issues.

We consider binary state variables in the main body of this paper. In general, the state need not be binary. For example, we could generalize the above example so which candidate would preform better on non-partisan issues is on a scale of 1 to 10 (where 1 indicates candidate $a$ is much better and 10 indicates candidate $b$ is much better). Predetermined voters' preferences would still not depend on the state. However, contingent voters' may have different thresholds on the state where they would transfer their support from candidate $a$ to candidate $b$. In Appendix E, we discuss the non-binary setting and extend our results to this setting.

The state variable is not directly observable in the election phase, as the performance of a new government official, a new hired employee, a new policy, etc., may not be revealed until many years after the vote. Instead, each voter receives a signal that is correlated to the hidden state which models the information voters received from difference sources.

Our goal is to select the *majority wish*, the alternative that would be preferred by the majority if they knew the state of the world. If some type of predetermined voters forms a majority, this is rather easy. However, in the case where the predetermined voters of neither alternative forms a majority, the mechanism needs to aggregate the information and preferences of the contingent voters and selects the alternative which, for a majority of voters, is "preferable for them".

## 1.2 Imperfectly Informed Voters

**Social Choice**  Social choice theory with imperfectly informed voters dates back to Condorcet's jury theorem in 1785 [De Condorcet, 1785], and has also been widely studied [Miller, 1986, Young, 1988, Ladha, 1992, Nitzan and Paroush, 2017]. In Condorcet's setting, there are two alternatives, one of which is "correct", and each voter votes for the correct alternative with probability $p$. Condorcet's jury theorem states that the probability that the majority voting scheme outputs the correct alternative goes to 1 as the number of voters increases when $p > 0.5$, and, conversely, this probability goes to 0 when $p < 0.5$.

Two unfortunate limitations for Condorcet's jury theorem are 1) It fails to output the correct alternative in the case voters' beliefs are aligned to the incorrect alternative (i.e., $p < 0.5$), and 2) It assumes voters vote truthfully and disregards voters' potential strategic behaviors. However, even in the case all voters have the same preference for the correct alternative, voting truthfully still may not be a Nash equilibrium [Austen-Smith and Banks, 1996].

To circumvent the first limitation, Feddersen and Pesendorfer [1997] consider the scenario where voters play a Nash equilibrium strategy profile, while the voting rule is still the majority scheme. Feddersen and Pesendorfer [1997] show that when the number of voters tends to infinity, the probability that a majority voting scheme outputs the correct alternative approaches to 1 if voters play the equilibrium strategy profile, while this probability is bounded away from 1 if voters play the truthful strategy profile instead. Feddersen and Pesendorfer's model assigns each voter a preference parameter $x \in [-1, 1]$ describing his/her alignment to the two alternatives. In their model, the unique (Bayes) Nash equilibrium is characterized by two thresholds $x_0, x_1 \in [-1, 1]$ with $x_0 < x_1$ such that voters with preferences below $x_0$ always vote for one alternative, voters with preferences above $x_1$ always vote for the other, and voters with preferences between $x_0$ and $x_1$ vote truthfully. Although Feddersen and Pesendorfer's solution guarantees that the correct alternative is output with high probability, it requires sophisticated voters. The voters need to calculate the values of $x_0$ and $x_1$ to decide their actions. The values $x_0$ and $x_1$ are each the zero point of a continuous monotone function involving a complicated Riemann integral. This is usually too demanding for voters in practice,

especially those who do not have a mathematical background. Moreover, it needs to be common knowledge that all agents can and will perform this computation.

In this paper, we take a different approach. Instead of asking voters to play the Nash equilibrium for majority voting, we seek to design a more sophisticated voting scheme, or a mechanism, than the majority voting scheme, such that voters are incentivized to vote *truthfully* under the mechanism, while guaranteeing the correct alternative is output with high probability. Our social choice mechanism thus elicits truthful information from the voters and then aggregates it.

**Information Aggregation**    The information aggregation literature considers how to obtain a "correct answer" by aggregating individuals' partial information—the crowd's wisdom. The straightforward procedure of outputting the answer that is believed to be correct by the majority does not always work [Chen et al., 2004, Simmons et al., 2011]. An example where this fails is when the crowd has a strong prior belief for the incorrect answer while novel specialized knowledge is only shared among a minority of the agents. It is also known that further calibration based on collecting participants' confidences (the posterior of their beliefs) does not always solve this problem [Hertwig, 2012, Prelec et al., 2017]. In a seminal work by Prelec et al. [2017], a new "surprisingly popular" approach was proposed: the participants' predictions over the other remaining participants' reported answers are collected, and the answer that is reported by more participants than predicted is output (we will review this in Section 3.1). They justified this approach both theoretically and through experimentation. In particular, they demonstrate the viability of approaches that require agents to predict reports of the other agents. Hosseini et al. [2021] empirically extend the surprisingly popular approach to the non-binary setting, where the goal is to learn the correct *ranking over many options* instead of the correct answer in two options.

The work by Prelec et al. [2017] and Hosseini et al. [2021] does not fit into the social choice context in two aspects. Firstly, the objective for an information aggregation mechanism is to output the correct answer. Participants who collaboratively contribute their knowledge/information do not have preferences on which answer is finally selected. This is fundamentally different from the social choice setting where the whole point of a social choice mechanism is to select an alternative favored by the majority. Secondly, as agents care about the outcome, agents may be strategic and manipulate their reports in order to make their preferred alternatives win, while Prelec et al. [2017] and Hosseini et al. [2021] do not put the problem in a game theory setting.

## 1.3   Our Results

In this paper, we study the social choice problem in a game theory setting with the existence of imperfectly informed voters who only have partial information regarding which alternative is more favorable. For various settings with two alternatives, we propose a mechanism that aggregates participants' private information and outputs the alternative favored by more than half of the participants—*the majority wish*. Our mechanisms are truthful, in the sense that the truthful strategy profile forms a *strong Bayes Nash Equilibrium*.

Our main wisdom-of-the-crowd-voting mechanism, presented in the paper, applies to the case of two worlds/states, where each agent receives a binary signal. We show it has strong truthfulness and aggregation properties, even for relatively small numbers of agents (Section 3). This result requires that the distribution of agents is a common knowledge. In the full version, we show several additional results. First, the common knowledge assumption is necessary to attain a strongly truthful mechanism that outputs the majority wish with high probability (Appendix D). Additionally, we extend our results to the setting of more than two worlds (Appendix E.2). Finally, we show how to extend our results to the case of more than two signals (Appendix E.3).

Our mechanism can easily be implemented using a simple questionnaire that elicits voters' information and preferences. The questions in the questionnaire are friendly to those voters who do not have relevant backgrounds in mathematics, game theory, etc., and they require participants to predict the responses of other agents, which is empirically validated by the surprisingly popular method from Prelec et al. [2017].

We ensure our mechanisms have a group truthfulness property by employing a "median trick." Intuitively, by the Median Voter Theorem [Black, 1948, Hotelling, 1929], the median voter's vote (in a binary choice) is favored by the majority. By a careful design, our mechanism ensures that

the voters who are "below" the median have a conflict of interest to the voters who are "above" the median, which makes sure less than half of the voters have an incentive to deviate and those voters can only change the outcome in the unfavorable direction by the property of median.

From a high level, our work can be understood as a revelation principal applied to plurality voting.[4] However this view is not entirely accurate. First, our equilibrium concept is strong Bayes Nash equilibrium while we only know that plurality voting implements the majority wish outcome in (Bayesian) Nash equilibrium [Feddersen and Pesendorfer, 1997]. Second, the revelation principal requires that agents report all their knowledge. In our case, this would include the entire prior, which is not realistic. In contrast, our mechanisms only require that agents report a preference and a prediction of other agents' preferences. Such reporting requirements have previously been shown to be pragmatic [John et al., 2012, Prelec et al., 2017, Hosseini et al., 2021]. Third, our setting is different than prior work [Feddersen and Pesendorfer, 1997], and this makes our results incomparable. In particular, we deal with a discrete state space. This difference also allows us to achieve some of our results not just in the limit, but for finite sets of agents.

Appendix A contains an additional comparison of our work with Feddersen and Pesendorfer [1997]'s.

### 1.4 Additional Related Work

Our work is additionally related to the recent work on incentive compatible machine learning [Perote and Perote-Pena, 2004, Dekel et al., 2010, Chen et al., 2018]. In these settings, the "social choice" being made is a machine learning predictor where agents benefit from their point having small error with respect to the chosen predictor. As in our setting, the information of the optimal model is distributed among the agents. Unlike our model, the private information and the preferences of the agents essentially coincide.

Information elicitation without verification, sometimes call peer prediction, is another very related line of research which shares some of the intuitions and techniques from information aggregation. The information elicitation literature has been well established in the past decades, starting from Prelec [2004]'s Baysian Truth Serum and Miller et al. [2005]'s peer-prediction method. These mechanisms cleverly design payments to the agents to guarantee that the truthful reporting of received information forms a Nash equilibrium.

A mass of recent work (see Faltings and Radanovic [2017] for a survey) is dedicated to designing information elicitation mechanisms that work in more general settings (such as, supporting a small number of agents [Dasgupta and Ghosh, 2013, Zhang and Chen, 2014, Schoenebeck and Yu, 2020], allowing agents having information with different levels of sophistication [Gao et al., 2016, Kong and Schoenebeck, 2018b]), or achieving better truthful guarantees (such as, strict Nash equilibrium [Schoenebeck and Yu, 2021], informed Nash equilibrium [Shnayder et al., 2016], or even dominant strategy equilibrium [Kong and Schoenebeck, 2019, Kong, 2020]) sometimes by studying more restrictive settings (e.g. multiple similar questions being asked simultaneously [Dasgupta and Ghosh, 2013]). Indeed, following Bayesian Truth Serum, many of these mechanism require the agents to predict other agents' reports [Zhang and Chen, 2014, Kong and Schoenebeck, 2018a,b, 2019, Kong et al., 2020, Schoenebeck and Yu, 2020].

However, all these mechanisms rely on *payments* to the agents to incentivize truth-telling. In our social choice setting, on the other hand, we need to incentivize truth-telling solely based on choosing the winning alternative.

## 2 Model and Preliminaries

In this paper, we will define our model and present our main result with two states and two signals. The extension to general numbers of states and signals is discussed in Appendix E. We use the standard signal structure that is commonly used in the peer prediction (e.g., the work in Sect. 1.4), information aggregation (e.g., [Prelec et al., 2017]), and social choice literature (e.g., [Feddersen and Pesendorfer, 1997]).

---

[4]Loosely speaking, the revelation principal states that any outcome that can be implemented in equilibrium can also be truthfully implemented in equilibrium by having the mechanism play the equilibrium strategy on behalf of the truthful agents.

Suppose a department of $T$ faculty members, or agents, need to decide whether or not to hire a new faculty candidate. In our model, those $T$ agents are voting for two *alternatives*, **A** and **R** (corresponding to "accept" and "reject"). There is a set of 2 possible *worlds* (or *states*) $\mathcal{W} = \{L, H\}$, which describes the underlying quality of the candidate. Here, $L$ stands for "low quality", and $H$ stands for "high quality". Agents do not know which world is the actual world that they are in. They have a common prior belief on the likelihood of each world. In the candidate hiring example, the CV of the candidate is given to those $T$ faculty members before any individual interviews, and a prior belief is formed. Let $W$ be the actual world which is viewed as a random variable. We use $W = L$ and $W = H$ to denote the events that the actual world is $L$ and $H$ respectively. Let $(P_L, P_H) = (\Pr(W = L), \Pr(W = H))$ be the prior over worlds. Each agent knows the values of $P_L$ and $P_H$ as prior beliefs. We further assume $P_L, P_H > 0$.

An individual interview for this candidate is held for each of the $T$ agents. Each agent $t$ receives a *signal*, represented by the random variable $S_t$, from the set $\mathcal{S} = \{\ell, h\}$. Given $W = L$ or $W = H$, the signals agents receive have the same distribution and are conditionally independent. Let $P_{\ell L} = \Pr(S_t = \ell \mid W = L)$ be the probability that signal $\ell$ will be received (by an arbitrary agent $t$) if the actual world is $L$. Let $P_{hL}, P_{\ell H}$ and $P_{hH}$ have similar meanings. The set of values $\{P_{\ell L}, P_{\ell H}, P_{hL}, P_{hH}\}$ is known by all the agents, but not the mechanism (such mechanisms are called *detail-free* in the peer prediction literature). Naturally, signals are positively correlated to the worlds:

$$P_{\ell L} > P_{\ell H} \qquad \text{and} \qquad P_{hH} > P_{hL}. \tag{1}$$

Different agents may have different bars on the quality of the candidate. For example, if a theory candidate is interviewed in a computer science department, the theory faculty members may want to accept this candidate at a lower quality compared with the AI, software and hardware faculty members. To capture this, each agent $t$ is assigned a *utility function* $v_t : \mathcal{W} \times \{\mathbf{A}, \mathbf{R}\} \to \{0, 1, \dots, B\}$. Naturally, voters receive higher utilities for **A** in world $H$ and for **R** in world $L$:

$$v_t(H, \mathbf{A}) > v_t(L, \mathbf{A}) \qquad \text{and} \qquad v_t(H, \mathbf{R}) < v_t(L, \mathbf{R}). \tag{2}$$

Since we can always rescale agents' utilities, for simplicity, we assume without loss of generality that agents' utilities are integers and bounded by $B \in \mathbb{Z}^+$. Endowed with their prior beliefs, upon receiving their signals, agents will have posterior beliefs about the distribution of $W$ and react to the mechanism in a way maximizing their expected utilities accordingly.

We assume $v_t(L, \mathbf{A}) \neq v_t(L, \mathbf{R})$ and $v_t(H, \mathbf{A}) \neq v_t(H, \mathbf{R})$ for each agent $t$, so that agents always strictly prefer one alternative over the other. Let $F$ be the set of the *candidate-friendly* agents $t$ who always prefer **A**: $v_t(H, \mathbf{A}) > v_t(L, \mathbf{A}) > v_t(L, \mathbf{R}) > v_t(H, \mathbf{R})$. Let $U$ be the set of the *candidate-unfriendly* agents $t$ who always prefer **R**: $v_t(L, \mathbf{R}) > v_t(H, \mathbf{R}) > v_t(H, \mathbf{A}) > v_t(L, \mathbf{A})$. Let $C$ be the set of the *contingent* agents $t$ whose preference depends on the actual world: $v_t(L, \mathbf{R}) > v_t(L, \mathbf{A})$ and $v_t(H, \mathbf{A}) > v_t(H, \mathbf{R})$.

Let $\alpha_F = \frac{|F|}{|T|}, \alpha_U = \frac{|U|}{|T|}$ and $\alpha_C = \frac{|C|}{|T|}$ be the fractions of the three types of agents. Since the numbers of theory, AI, software, hardware faculty members are known to everyone, we assume that the values of $\alpha_F, \alpha_U$ and $\alpha_C$ are common knowledge. Admittedly, this assumption may not apply to some specific scenarios. In Appendix D, we discuss the model where agents have only partial information on $\alpha_F, \alpha_U$ and $\alpha_C$, and present a strong impossibility result for this model.

The goal is to output the *majority wish*, the alternative that is preferred by at least half of the agents conditioned on the true state. We assume $T$ is an odd number to avoid ties. Clearly, **A** should be output if $\alpha_F > \frac{1}{2}$, **R** should be output if $\alpha_U > \frac{1}{2}$. In the case $\alpha_F, \alpha_U < \frac{1}{2}$, **A** should be output if the actual world is $H$ and **R** should be output if the actual world is $L$.

Our results will sometimes require $T$, the number of agents, to be sufficiently large, and it may be helpful to think of $T \to \infty$. However, we will always assume that the parameters of the model: $B$, $\{P_L, P_H\}$, $\{P_{\ell L}, P_{\ell H}, P_{hL}, P_{hH}\}$, and $\{\alpha_F, \alpha_U, \alpha_C\}$, do not depend on $T$ in any way.

The traditional social choice setting with agents having predetermined preferences can be viewed as a special case of our model, by setting $|C| = 0$ (i.e., there is no contingent agent).

For the ease of comprehension, we have used the faculty candidate hiring as a running example for this paper. This can be replaced by any example from most practical scenarios where different types of imperfectly informed voters are voting between two alternatives, including all the examples we mentioned in Section 1.

In our election example in the second paragraph of Section 1, **A** and **R** can represent candidates $a$ and $b$ respectively. Correspondingly, for this example, $L$ and $H$ can represent "$a$ is better suited" and "$b$ is better suited" respectively. $F$ and $U$ represent voters aligned to party $A$ and $B$ respectively, while $C$ represents those swing voters whose preferences depend on the signals (in this case, the signals correspond to their private experiences and beliefs, which may be based on information they obtained from their favorite TV programs, newspapers, etc).

## 2.1 Strategy and $\varepsilon$-strong Bayes Nash equilibrium

A *mechanism* collects a *report* from each agent, and then outputs an alternative which is either **A** or **R**. The mechanism specifies the content of the report by specifying questions for the agents. Examples of those questions include asking each agent for the signal he/she receives, asking each agent to predict the other agents' reports, etc.

Let $\mathcal{R}$ be the space of all possible reports, which depends on the design of the mechanism. A *pure strategy* of an agent is given by a function $\sigma : \mathcal{S} \to \mathcal{R}$ that maps a signal received by this agent to a report. In a *mixed strategy*, $\sigma$ can be a random function.

An agent's strategy is *truthful* if it always specifies the correct answer to each question in the report, to the best of the agent's knowledge after receiving the signal. For example, if the mechanism asks for the agent's signal, an agent playing the truthful strategy should report the signal he/she receives; if the mechanism asks the agents to predict the fraction of agents who will receive signal $m$, an agent playing the truthful strategy should report his/her posterior belief on this computed by the Bayes rule (Section 2.2 discusses the computation of posterior beliefs).

Given a strategy profile $\Sigma = (\sigma_1, \ldots, \sigma_T)$, let $u_t(\Sigma)$ be the expected utility of agent $t$, where the expectation is taken over the sampling of agents' signals. Notice that we use $u_t$ to denote the *ex-ante* utility (as defined just now) and we have used $v_t$ to denote the *ex-post* utility (see the fourth paragraph in Section 2). Most parts of this paper will focus on the *ex-ante* utility, especially when we are talking about any equilibrium solution concept.[5]

Since we are in a social choice setting with a potentially large number of agents, a single agent's behavior may not have much effect. Thus, instead of the typical Bayes Nash Equilibrium, we consider a much stronger goal—the strong Bayes Nash equilibrium.

**Definition 2.1.** A strategy profile $(\sigma_1, \ldots, \sigma_T)$ is an $\varepsilon$-*strong Bayes Nash equilibrium* if there does not exist a subset of agents $D$ and a strategy profile $(\sigma'_1, \ldots, \sigma'_T)$ such that

1. $\sigma_t = \sigma'_t$ for each $t \notin D$,

2. $u_t(\sigma'_1, \ldots, \sigma'_T) \geq u_t(\sigma_1, \ldots, \sigma_T)$ for each $t \in D$, and

3. there exist $t \in D$ such that $u_t(\sigma'_1, \ldots, \sigma'_T) > u_t(\sigma_1, \ldots, \sigma_T) + \varepsilon$.

When a strategy profile $(\sigma_1, \ldots, \sigma_n)$ is not an $\varepsilon$-strong Bayes Nash equilibrium, we will call the subset of the agents $D$ in Definition 2.1 the *deviating agents* or the *deviating coalition*. By Definition 2.1, every agent of the deviating coalition must be at least as well off, and some must be strictly better off by at least $\epsilon$. Notice that a 0-strong Bayes Nash equilibrium is the conventional strong Bayes Nash equilibrium. The larger $\varepsilon$ is, the harder it is to find a deviating coalition, and so the larger the set of $\varepsilon$-strong Bayes Nash equilibria.

## 2.2 Posterior Update by Bayes Rule

Upon receiving a signal $S_t \in \{\ell, h\}$, agent $t$ updates his/her posterior beliefs (about the probability that (s)he is in world $L$ or $H$, the fraction of agents that will receive signal $\ell$ or $h$, etc.) based on

---

[5]However, the goal of the mechanism is to output the majority preferred alternative based on agents' *ex-post* utilities. Our choice of discussing *ex-ante* utilities in solution concept and *ex-post* utilities in the goal is motivated by practical applications. The underlying better alternative is unknown in the election phrase, so *ex-ante* utilities are considered for the solution concept. On the other hand, the ultimate goal is to select the alternative that *turns out to be better* after a certain period of time when the underlying better alternative is revealed (e.g., the performance of a selected president or the effect of a new policy in the next few years after the election), and in this case the *ex-post* utilities are considered.

Bayes rule. Let $T_{m'm}$ be the probability that an agent who receives signal $m \in \{\ell, h\}$ believes that another agent will receive signal $m' \in \{\ell, h\}$. Straightforward calculations by Bayes rule (available in Appendix B.1) reveals

$$T_{m'm} = \frac{P_L P_{mL}}{P_L P_{mL} + P_H P_{mH}} \cdot P_{m'L} + \frac{P_H P_{mH}}{P_L P_{mL} + P_H P_{mH}} \cdot P_{m'H}. \tag{3}$$

Given a strategy profile $\Sigma = \{\sigma_1, \ldots, \sigma_T\}$ and a mechanism $\mathcal{M}$, let $\lambda_n^{\mathbf{A},\mathcal{M}}(\Sigma)$ be the probability that alternative $\mathbf{A}$ is announced as the winner given the actual world is $n$, then $\lambda_n^{\mathbf{R},\mathcal{M}}(\Sigma) = 1 - \lambda_n^{\mathbf{A},\mathcal{M}}(\Sigma)$ is the probability that alternative $\mathbf{R}$ wins given the actual world is $n$. We will omit the superscript $\mathcal{M}$ when it is clear what mechanism we are discussing.

All the agents' *ex-ante* utilities depend exclusively on $\lambda_L^{\mathbf{A}}(\Sigma), \lambda_H^{\mathbf{A}}(\Sigma)$ (or $\lambda_L^{\mathbf{R}}(\Sigma), \lambda_H^{\mathbf{R}}(\Sigma)$), and each agent $t$'s utility is given by

$$u_t(\Sigma) = P_L \left( \lambda_L^{\mathbf{A}}(\Sigma) v_t(L, \mathbf{A}) + \lambda_L^{\mathbf{R}}(\Sigma) v_t(L, \mathbf{R}) \right) + P_H \left( \lambda_H^{\mathbf{A}}(\Sigma) v_t(H, \mathbf{A}) + \lambda_H^{\mathbf{R}}(\Sigma) v_t(H, \mathbf{R}) \right), \tag{4}$$

which can also be rewritten as (by noticing $\lambda_L^{\mathbf{A}}(\Sigma) + \lambda_L^{\mathbf{R}}(\Sigma) = 1$ and $\lambda_H^{\mathbf{A}}(\Sigma) + \lambda_H^{\mathbf{R}}(\Sigma) = 1$)

$$\begin{aligned} u_t(\Sigma) = &P_L v_t(L, \mathbf{R}) + P_H v_t(H, \mathbf{R}) + P_L \lambda_L^{\mathbf{A}}(\Sigma)(v_t(L, \mathbf{A}) - v_t(L, \mathbf{R})) \\ &+ P_H \lambda_H^{\mathbf{A}}(\Sigma)(v_t(H, \mathbf{A}) - v_t(H, \mathbf{R})). \end{aligned} \tag{5}$$

We will always use $\Sigma^* = \{\sigma_1^*, \ldots, \sigma_T^*\}$ to denote the truthful strategy profile.

# 3 The Wisdom-of-the-Crowd-Voting Mechanism

We will first review Prelec et al.'s *Surprisingly Popular* algorithm [Prelec et al., 2017], which works under a setting similar to ours but with non-strategic agents. Some part of the intuition behind our mechanism is based on Prelec et al.'s work.

## 3.1 Prelec et al.'s Surprisingly Popular Algorithm

For the purpose of this paper, we will describe the algorithm with two worlds and two signals. The algorithm asks each agent $t$ the signal (s)he receives, and his/her belief on the fraction of agents who have received signal $\ell$ (or signal $h$). In our notation, each agent reports the realization of $S_t$ and, assuming $S_t = m \in \{\ell, h\}$, the value of $T_{\ell m}$ (or $T_{hm}$, which equals to $1 - T_{\ell m}$). Since agents are assumed to be non-strategic, those who receive signal $\ell$ will report $(\ell, T_{\ell\ell})$ and those who receive signal $h$ will report $(h, T_{\ell h})$. The algorithm then computes the fraction of agents who report signal $\ell$, and the average value of all the reported $T_{\ell m}$'s. If the former is greater than the latter, $\ell$ is considered as being "surprisingly popular" and the algorithm will conclude that $L$ is the actual world. Otherwise, $h$ will be considered as being "surprisingly popular" and $H$ will be concluded as being the actual world.

The correctness of this algorithm is based on the following simple yet important observation in Theorem 3.1. In particular, the average of agents' reported predictions (those $T_{\ell m}$'s) will be between $T_{\ell h}$ and $T_{\ell\ell}$. When the number of agents $T$ is sufficiently large, the actual fraction of agents who receive signal $\ell$ will be either approximately $P_{\ell L}$ (if $L$ is the actual world) or approximately $P_{\ell H}$ (if $H$ is the actual world). Theorem 3.1 then implies the correctness of the Surprisingly Popular algorithm.

**Theorem 3.1.** $P_{\ell H} < T_{\ell h} < T_{\ell\ell} < P_{\ell L}$ and $P_{hH} > T_{hh} > T_{h\ell} > P_{hL}$

For completeness, the full proof is appears in Appendix B.2, but the intuition is straightforward. The inequality $P_{\ell H} < P_{\ell L}$ is by the positive correlation (1). The inequality $T_{\ell h} < T_{\ell\ell}$ is also intuitive: the positive correlation between the signals and worlds implies the positive correlation between two agents' received signals. Finally, (3) implies each of $T_{\ell h}$ and $T_{\ell\ell}$ is a weighted average of $P_{\ell H}$ and $P_{\ell L}$, so the value is between $P_{\ell H}$ and $P_{\ell L}$. This concludes the first inequality chain, and the second can be shown similarly.

Throughout this section, we use $c$ to denote the following constant, which is used in the Chernoff bounds in our proofs.

$$c = \frac{1}{3} \min \{ T_{\ell h} - P_{\ell H}, T_{\ell\ell} - T_{\ell h}, P_{\ell L} - T_{\ell\ell}, P_{hH} - T_{hh}, T_{hh} - T_{\ell h}, T_{\ell h} - P_{\ell H} \} \tag{6}$$

## 3.2 The Wisdom-of-the-Crowd-Voting Mechanism

---

**Mechanism 1** The Wisdom-of-the-Crowd-Voting Mechanism

---
1: Each agent $t$ reports his/her type ($F$, $U$ or $C$) to the mechanism, and if he/she is of type $C$, the signal (s)he receives (either $\ell$ or $h$), denoted by $\bar{s}_i \in \{\ell, h\}$.
2: If agent $t$ reports type $F$, his reported signal will be automatically treated as $\bar{s}_t = h$; if agent $t$ reports type $U$, his reported signal will be automatically treated as $\bar{s}_t = \ell$.
3: For each agent $t$ of type $C$, ask him/her to predict the fraction of agents who will report signal $h$. Let $\bar{\delta}_t$ be $t$'s prediction. *The prediction $\bar{\delta}_t$ should be made with the type $F$ and type $C$ agents' predictions defined in the previous step being considered, and the mechanism makes this clear to the agents.* For each agent $t$ of type $F$, $\bar{\delta}_t$ is set to 0, and for each agent $t$ of type $U$, $\bar{\delta}_t$ is set to 1.
4: Compute the *median* of those $\bar{\delta}_t$, denoted by $\bar{\delta}$.
5: If more than half of the agents report type $F$, announce **A** being the winning alternative; if more than half of the agents reports type $U$, announce **R** being the winning alternative.
6: If the fraction of the agents reporting $\bar{s}_t = h$ is more than the median $\bar{\delta}$, announce **A** being the winning alternative; otherwise, announce **R** being the winning alternative.

---

At Step 3 of the mechanism, we only elicit predictions from those contingent agents, and those candidate-friendly (candidate-unfriendly resp.) agents' predictions are treated as 0 (1 resp.). In Appendix C, we discuss an alternative mechanism where we elicit all the predictions and then take the median. The alternative mechanism shares the same theoretical properties, and we discuss the advantages and disadvantages of this alternative.

Mechanism 1 may look too obscure to be implemented in practice. However, very simple and understandable questionnaires implementing the mechanism can be designed. In our running example of faculty candidate hiring, the questionnaire corresponding to our mechanism could look like:

1. Choose one of the followings:

   (a) I definitely want to accept this candidate, regardless of my colleagues' inputs.
   (b) I definitely want to reject this candidate, regardless of my colleagues' inputs.
   (c) After talking to the candidate, I am more inclined to accept him/her than before.
   (d) After talking to the candidate, I am more inclined to reject him/her than before.

2. If your answer is (c) or (d) in the first question, what percentage of the faculty members do you believe will choose (a) or (c) in the first question?

In the first question above, faculty members of type $F$ (type $U$ resp.) will choose (a) ((b) resp.). Faculty members of type $C$ will choose either (c) or (d) depending on the signals they have received. If a faculty member receives signal $h$, (s)he believes world $H$ is more likely than before. Notice that, it is still possible that (s)he believes the probability of world $H$ being the actual world is less than $50\%$ and (s)he still prefers rejecting the candidate based on his/her private information (for example, his/her prior belief may be only $10\%$ for world $H$, and his/her posterior belief for this increases to $30\%$ upon receiving a signal $h$), so the description that (s)he is "more inclined to accept the candidate than before" accurately implements Mechanism 1. The same holds for those receiving signal $\ell$.

## 3.3 Main Theoretical Results for Our Mechanism

We first show that our mechanism indeed achieves (with an exponentially small failure probability) the goal of outputting the alternative favored by the majority, assuming agents are truth-telling.

**Theorem 3.2.** *If all the agents play the truthful strategy $\Sigma^*$, then, with probability at least $1 - 2\exp(-2c^2\alpha_C T)$ (where c is the constant defined in Eqn. (6)), our mechanism outputs an alternative that is favored by more than half of the agents.*

*Proof.* Suppose all the agents report truthfully. Step 5 of the mechanism guarantees that the majority wish will be announced with probability 1 if either $\alpha_F > 0.5$ or $\alpha_U > 0.5$. It remains to consider the case where we have both $\alpha_F < 0.5$ and $\alpha_U < 0.5$.[6] In this case, **A** is favored by the majority if the actual world is $H$, and **R** is favored by the majority if the actual world is $L$.

---

[6]Recall that we have assume $T$ is an odd number, so we cannot have $\alpha_F = 0.5$ or $\alpha_U = 0.5$.

If a contingent agent $t$ receives signal $h$, (s)he will believe that a $T_{hh}$ fraction of agents receive $h$ (before the treatment at Step 2), and (s)he will report $\bar{\delta}_t = \alpha_C T_{hh} + \alpha_F$ (after considering the treatment at Step 2). Similarly, a contingent agent receiving signal $\ell$ will report $\bar{\delta}_t = \alpha_C T_{h\ell} + \alpha_F$. We have $\bar{\delta}_t = 0$ for each candidate-friendly agent and $\bar{\delta}_t = 1$ for each candidate-unfriendly agent. Since we are considering the case $\alpha_F, \alpha_U < 0.5$, the median $\bar{\delta}$ is in the interval $[\alpha_C T_{h\ell} + \alpha_F, \alpha_C T_{hh} + \alpha_F]$ (note that $T_{h\ell} < T_{hh}$ by Theorem 3.1).

Suppose the actual world is $L$. The expected fraction of the agents receiving signal $h$ would be $P_{hL}$, and the expected fraction of the agents reporting signal $h$ (after the treatment at Step 2) would be $\alpha_C \cdot P_{hL} + \alpha_F$. By a Chernoff bound, with probability at least $1 - 2\exp(-2c^2\alpha_C T)$, the fraction of agents reporting signal $h$ is in the interval $[\alpha_C \cdot (P_{hL} - c) + \alpha_F, \alpha_C \cdot (P_{hL} + c) + \alpha_F]$, which is less than $\alpha_C \cdot T_{h\ell} + \alpha_F \leq \bar{\delta}$ by Theorem 3.1 and (6). Step 6 of the mechanism indicates that $\mathbf{R}$ will be announced. The analysis for the case where $H$ is the actual world is similar. $\qquad\square$

Next, we show that the truthful strategy profile is an $\varepsilon$-strong Bayes Nash equilibrium of our mechanism for some exponentially small $\varepsilon$.

**Theorem 3.3.** *The truthful strategy profile is an $\varepsilon$-strong Bayes Nash equilibrium, where $\varepsilon = (2B^2 + 4B)\exp(-2c^2\alpha_C T)$.*

## 3.4 Proof Theorem 3.3 with $T \to \infty$

We defer the full proof of Theorem 3.3 to Appendix B.3. Here we prove the following limit version, where $T \to \infty$, which illustrates the key features while eliminating the need for both Chernoff bound analyses and for some additional subtle corner cases.

**Theorem 3.4.** *For $T \to \infty$, the truthful strategy profile is a strong Bayes Nash equilibrium.*

We consider three cases: 1) $\alpha_F > 0.5$, 2) $\alpha_U > 0.5$ and 3) $\alpha_F < 0.5$ and $\alpha_U < 0.5$.

In the first case, more than half of the agents are candidate-friendly, and $\mathbf{A}$ will be announced according to Step 5 of the mechanism if these agents report truthfully. The truthful strategy profile forms a strong Bayes Nash equilibrium, as those candidate-friendly agents receive their maximum utilities by truth-telling and the remaining agents are not able to stop the mechanism from outputting $\mathbf{A}$ regardless of what they report. The analysis for the second case is the analogous to the first case. It remains to consider the third case.

Under the third case, $\mathbf{A}$ is favored by the majority if the actual world is $H$, and $\mathbf{R}$ is favored by the majority if the actual world is $L$. By the same analysis as in the proof of Theorem 3.2, supposing agents report truthfully, we know that $\mathbf{A}$ will be output with probability 1 (by taking the limit $T \to \infty$) if the actual world is $H$, and $\mathbf{R}$ will be output if the actual world is $L$. Therefore, the contingent agents (type $C$) receive their maximum utilities, and thus have no incentive to deviate from the truthful strategy.

To conclude that truth-telling is a strong Bayes Nash equilibrium, we will show that there is no coalition of deviating agents $D$ (see the paragraph following Definition 2.1). Let $\Sigma'$ be the strategy profile after $D$'s deviation.

Next, we show that $D$ cannot contain both a type $F$ agent and a type $U$ agent. The following lemma shows that an increase in a type $F$ agent's (*ex-ante*) utility always results a decrease in a type $U$ agent's (*ex-ante*) utility, and vice versa. This is obvious if we are dealing with *ex-post* utilities, as a candidate-friendly agent and a candidate-unfriendly agent always want the opposite alternatives. However, this becomes less obvious for *ex-ante* utilities.

**Lemma 3.5.** *Suppose $\alpha_F < 0.5$ and $\alpha_U < 0.5$. Let $\Sigma^*$ be the truthful strategy profile and $\Sigma'$ be an arbitrary strategy profile. Let $t_1$ be an arbitrary candidate-friendly agent and $t_2$ be an arbitrary candidate-unfriendly agent. We have*

**(i)** *If $u_{t_1}(\Sigma') - u_{t_1}(\Sigma^*) > 0$, then $u_{t_2}(\Sigma') - u_{t_2}(\Sigma^*) < 0$.*

**(ii)** *If $u_{t_2}(\Sigma') - u_{t_2}(\Sigma^*) > 0$, then $u_{t_1}(\Sigma') - u_{t_1}(\Sigma^*) < 0$.*

*Proof.* By (5), we have
$$u_t(\Sigma') - u_t(\Sigma^*) = \Gamma_L(v_t(L, \mathbf{A}) - v_t(L, \mathbf{R})) - \Gamma_H(v_t(H, \mathbf{A}) - v_t(H, \mathbf{R})),$$

where $\Gamma_L = P_L(\lambda_L^{\mathbf{A}}(\Sigma') - \lambda_L^{\mathbf{A}}(\Sigma^*))$ and $\Gamma_H = P_H(\lambda_H^{\mathbf{A}}(\Sigma^*) - \lambda_H^{\mathbf{A}}(\Sigma'))$. Theorem 3.2 implies $\lambda_L^{\mathbf{A}}(\Sigma^*) = 0$ and $\lambda_H^{\mathbf{A}}(\Sigma^*) = 1$ (with $T \to \infty$), which implies $\Gamma_L, \Gamma_H \geq 0$. By (2), we have $v_t(L, \mathbf{A}) - v_t(H, \mathbf{A}) < 0 < v_t(L, \mathbf{R}) - v_t(H, \mathbf{R})$, which further implies $v_t(L, \mathbf{A}) - v_t(L, \mathbf{R}) < v_t(H, \mathbf{A}) - v_t(H, \mathbf{R})$. Since a type $F$ agent always prefers $\mathbf{A}$ and a type $U$ agent always prefers $\mathbf{R}$,

$$0 < v_{t_1}(L, \mathbf{A}) - v_{t_1}(L, \mathbf{R}) \quad < \quad v_{t_1}(H, \mathbf{A}) - v_{t_1}(H, \mathbf{R}), \qquad \text{and}$$
$$v_{t_2}(L, \mathbf{A}) - v_{t_2}(L, \mathbf{R}) \quad < \quad v_{t_2}(H, \mathbf{A}) - v_{t_2}(H, \mathbf{R}) < 0.$$

Intuitively, this means $u_{t_1}(\Sigma') - u_{t_1}(\Sigma^*)$ is more sensitive to $\Gamma_H$ and $u_{t_2}(\Sigma') - u_{t_2}(\Sigma^*)$ is more sensitively to $\Gamma_L$. Formally, $u_{t_1}(\Sigma') - u_{t_1}(\Sigma^*) > 0$ implies $\Gamma_L > \Gamma_H \frac{v_{t_1}(H,\mathbf{A}) - v_{t_1}(H,\mathbf{R})}{v_{t_1}(L,\mathbf{A}) - v_{t_1}(L,\mathbf{R})} \geq \Gamma_H$, and $u_{t_2}(\Sigma') - u_{t_2}(\Sigma^*) \geq 0$ implies $\Gamma_L \leq \Gamma_H \frac{v_{t_2}(H,\mathbf{A}) - v_{t_2}(H,\mathbf{R})}{v_{t_2}(L,\mathbf{A}) - v_{t_2}(L,\mathbf{R})} \leq \Gamma_H$. Thus, $u_{t_1}(\Sigma') - u_{t_1}(\Sigma^*) > 0$ and $u_{t_2}(\Sigma') - u_{t_2}(\Sigma^*) \geq 0$ cannot be both true. The proof for (ii) is similar. $\qquad\square$

We have seen that the deviating coalition $D$ cannot contain any contingent agents (since their *ex-ante* utilities have already been maximized). Thus, Lemma 3.5 implies $D$ can only be comprised of either candidate-friendly agents or candidate-unfriendly agents. Finally, we show that a minority coalition comprised of only candidate-friendly agents or only candidate-unfriendly agents cannot change the outcome by misreporting. We consider candidate-friendly agents without loss of generality.

Suppose $D$ contains only candidate-friendly agents. Those agents cannot make the mechanism output $\mathbf{A}$ at Step 5, because fewer than $1/2$ of the agents report type $F$ no matter how agents in $D$ deviate. To maximize the chance for the mechanism to output $\mathbf{A}$ at Step 6, those candidate-friendly agents would like to maximize the fraction of agents reporting $h$ and minimize the median of the prediction $\bar{\delta}$. However, the mechanism already does this when those candidate-friendly agents play the truthful strategy: their signals are treated as $h$ at Step 2, and their predictions are treated as 0 at Step 3. The same analysis works when $D$ contains only candidate-unfriendly agents. This concludes the proof for Theorem 3.4.

In fact, the arguments in the previous paragraph show that: *truth-telling is a dominant strategy for each candidate-friendly agent and each candidate-unfriendly agent*.

## 4    Remarks, Limitations and Future Work

We have assumed agents are Bayesian. Although this assumption may not be completely realistic in practice, we believe the theoretical properties of our mechanism still hold to a certain extent. For example, agents may not exactly predict $\alpha_C T_{hh} + \alpha_F$ or $\alpha_C T_{h\ell} + \alpha_F$ in practice, but it is reasonable to assume that their predictions are roughly around these two numbers, or in between. If so, all the theoretical properties will still hold. Prelec et al. [2017] also assume Bayesian agents in the theoretical analysis of the surprisingly popular method, but their empirical experiments with human subjects suggest the method still works in practice.

Our mechanism can be extended to the setting where a fixed fraction $\tau$ of acceptance votes is required to adopt a policy. For example, in many countries, constitutional amendments requires a $2/3$ majority to pass. To do this, we only need to change Step 4 of Mechanism 1 such that $\bar{\delta}$ is the prediction with rank $\tau T$, and change Step 5 such that $\mathbf{A}$ is announced if more than $\tau$ fraction of agents report type $F$ and $\mathbf{R}$ is announced if more than $1 - \tau$ fraction of agents report type $U$.

One limitation is that our mechanism only deals with two alternatives. While this is natural in many scenarios (accept/reject, election with two candidates), extending our results to more than two alternatives is an interesting future direction, but faces a multitude of hurdles: the median technique will not straightforwardly work, the "surprisingly popular" formalism faces impossibility results [Prelec et al., 2017], and Gibbard-Satterthwaite social choice impossibility results apply.

Another interesting future direction is deployment. Our Wisdom-of-the-Crowd-Voting mechanism could be used in any place that currently uses majority voting to better aggregate information. Of course, it would suffer from some of the same drawbacks of majority voting: that the majority can impose their will on the minority. It is not clear if either one of these enjoys fairness properties not included by the other, but that would be another direction of future inquiry. It would be interesting to test this mechanism in the real world, and then test to see if participants are, in aggregate, more pleased with the alternatives selected by this mechanism as compared with those selected by a majority vote.

## Acknowledgement

We gratefully thank Noah Burrell and the anonymous reviewers for the helpful comments and suggestions for improving this paper.

## Funding Transparency Statement

All the funding support related to this paper has been declared at the bottom of Page 1.

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
