## A   Comparison with Feddersen and Pesendorfer's Work

Feddersen and Pesendorfer [1997] consider a two-alternative setting similar to our model. As mentioned before, Feddersen and Pesendorfer [1997] consider the standard majority voting where each agent votes for an alternative, while assuming agents play a (Bayes) Nash equilibrium strategy profile. We, on the other hand, design a more sophisticated mechanism to incentivize truth-telling.

Other than this difference, the state space, the signal space and the space of agents' types in Feddersen and Pesendorfer [1997] are all continuous. For Feddersen and Pesendorfer's continuous setting, in the Nash equilibrium, agents' strategies have three types: always vote for one alternative, always vote for the other, and vote the alternative based on the signal. These are similar to our three types: candidate-friendly, contingent, candidate-unfriendly. However, due to continuity, each agent needs to compute his/her type by solving an equation with a complicated Riemann integral (while agents' know their types directly according to their utility functions in our setting). A phenomenon in their setting due to continuity is that the fraction of contingent voters in the Nash equilibrium approaches zero when the number of the voters goes to infinity.

Although agents can be classified by three types in both settings, we would like to clarify a fundamental difference in the motivation behind this classification. In our setting, each agent's type reflects his/her preference over the two alternatives. In Feddersen and Pesendorfer's setting, each agent "chooses" a type in a specific way so that the majority voting scheme outputs the correct alternative with high probability. Therefore, in their setting, agents' types are chosen for collaboratively aggregating information, and should not be viewed as reflections of their preferences. Although an agent's preference does affect his/her choice, the purpose for choosing a type is for information aggregation, not for reflecting the preference.

At a high level, our mechanism includes some novel techniques, including the surprisingly popular technique and the median trick, to ensure the output of the correct alternative in the setting with strategic agents. In Feddersen and Pesendorfer's setting, it may be surprisingly that the simple majority voting scheme is already enough for output the correct alternative. The reason behind this is that certain implicit techniques for guaranteeing the correct output are "embedded" into agents' strategic behaviors. In other words, the agents are the ones who work out those techniques, not the mechanism. That is why we mentioned in the introduction that the agents in Feddersen and Pesendorfer's setting need to have much more sophistication compared with the agents in our setting.

Another difference is that they are considering a Nash equilibrium strategy profile, while our mechanism satisfies the much stronger criterion that truth-telling strategies form a *strong* Bayes Nash equilibrium.

## B   Omitted Proofs

Equations, theorems, lemmas or propositions are restated for the ease of reading.

### B.1   Proof of Equation (3)

Equation (3) is restated below:

$$T_{m'm} = \frac{P_L P_{mL}}{P_L P_{mL} + P_H P_{mH}} \cdot P_{m'L} + \frac{P_H P_{mH}}{P_L P_{mL} + P_H P_{mH}} \cdot P_{m'H}.$$

*Proof.* Suppose agent $t$ receives signal $S_t = m \in \{\ell, h\}$. (S)he believes that the actual world is $n \in \{L, H\}$ with probability

$$\Pr\left(W = n \mid S_t = m\right) = \frac{\Pr(W = n)\Pr(S_t = m \mid W = n)}{\Pr(S_t = m)} = \frac{P_n P_{mn}}{P_L P_{mL} + P_H P_{mH}}.$$

Now, we can calculate $T_{m'm}$:

$$T_{m'm} = P_{m'L} \cdot \Pr\left(W = L \mid S_t = m\right) + P_{m'H} \cdot \Pr\left(W = H \mid S_t = m\right)$$

$$= \frac{P_L P_{mL}}{P_L P_{mL} + P_H P_{mH}} \cdot P_{m'L} + \frac{P_H P_{mH}}{P_L P_{mL} + P_H P_{mH}} \cdot P_{m'H}.$$

$\square$

## B.2 Proof of Theorem 3.1

**Theorem 3.1.** $P_{\ell H} < T_{\ell h} < T_{\ell \ell} < P_{\ell L}$ and $P_{hH} > T_{hh} > T_{h\ell} > P_{hL}$

*Proof.* We will only show the first inequality chain. The second chain follows directly from the first by noticing each term in the second chain is 1 minus a term in the first.

By (3) and $P_{\ell H} < P_{\ell L}$ in (1), we have

$$T_{\ell \ell} = \frac{P_L P_{\ell L}^2 + P_H P_{\ell H}^2}{P_L P_{\ell L} + P_H P_{\ell H}} < \frac{P_L P_{\ell L}^2 + P_H P_{\ell H} P_{\ell L}}{P_L P_{\ell L} + P_H P_{\ell H}} = P_{\ell L}$$

and

$$T_{\ell h} = \frac{P_L P_{\ell L} P_{hL} + P_H P_{\ell H} P_{hH}}{P_L P_{hL} + P_H P_{hH}} > \frac{P_L P_{\ell H} P_{hL} + P_H P_{\ell H} P_{hH}}{P_L P_{hL} + P_H P_{hH}} = P_{\ell H}.$$

Finally, to show $T_{\ell \ell} > T_{\ell h}$, it suffices to show that

$$\pi_1 := \frac{P_L P_{\ell L}}{P_L P_{\ell L} + P_H P_{\ell H}} > \pi_2 := \frac{P_L P_{hL}}{P_L P_{hL} + P_H P_{hH}},$$

since $T_{\ell \ell} = \pi_1 P_{\ell L} + (1 - \pi_1) P_{\ell H}$, $T_{\ell h} = \pi_2 P_{\ell L} + (1 - \pi_2) P_{\ell H}$ and $P_{\ell L} > P_{\ell H}$. Simple calculations show this:

$$\pi_1 > \frac{P_L}{P_L + P_H} > \pi_2,$$

where the first inequality is due to $P_{\ell L} > P_{\ell H}$ and the second inequality is due to $P_{hH} > P_{hL}$. $\square$

## B.3 Proof of Theorem 3.3

**Theorem 3.3.** *The truthful strategy profile is an $\varepsilon$-strong Bayes Nash equilibrium, where $\varepsilon = (2B^2 + 4B) \exp(-2c^2 \alpha_C T)$.*

### B.3.1 Proof Sketch for Theorem 3.3

We first describe a sketch of the proof. We consider three cases: 1) $\alpha_F > 0.5$, 2) $\alpha_U > 0.5$ and 3) $\alpha_F < 0.5$ and $\alpha_U < 0.5$.

**Case 1) $\alpha_F > 0.5$.** For the first case, more than half of the agents are candidate-friendly, and **A** will be announced according to Step 5 of the mechanism if these agents report truthfully. The truthful strategy profile forms a (0-)strong Bayes Nash equilibrium, as those candidate-friendly agents receive their maximum utilities by truth-telling and the remaining agents are not able to stop the mechanism from outputting **A** regardless of what they report.

**Case 2) $\alpha_U > 0.5$.** The analysis for the second case is the analogous to the first.

**Case 3) $\alpha_F < 0.5$ and $\alpha_U < 0.5$.** Under the third case, **A** is favored by the majority if the actual world is $H$, and **R** is favored by the majority if the actual world is $L$. Given a strategy profile $\Sigma$, we define

$$I(\Sigma) \equiv P_L \lambda_L^{\mathbf{A}}(\Sigma) + P_H \lambda_H^{\mathbf{R}}(\Sigma)$$

to be the error rate of a strategy: the probability the mechanism selects the alternative that is not the majority wish.

By Theorem 3.2, we know that $I(\Sigma^*) \leq 2 \exp(-2c^2 \alpha_C T)$. Specifically, supposing agents report truthfully, we know that **A** will be output with probability at least $1 - 2 \exp(-2c^2 \alpha_C T)$ if the actual world is $H$ (i.e., $\lambda_H^{\mathbf{A}}(\Sigma^*) \geq 1 - 2 \exp(-2c^2 \alpha_C T)$), and **R** will be output with probability at least $1 - 2 \exp(-2c^2 \alpha_C T)$ if the actual world is $L$ (i.e., $\lambda_L^{\mathbf{R}}(\Sigma^*) \geq 1 - 2 \exp(-2c^2 \alpha_C T)$).

To conclude that truth-telling is an $\varepsilon$-strong Bayes Nash equilibrium, we will study two cases and show that no coalition of deviating agents $D$ exists in either case. Recall that in a deviating coalition all agents must benefit and some agent must benefit by at least $\varepsilon = (2B^2 + 4B) \exp(-2c^2 \alpha_C T)$. Let $\Sigma'$ be the strategy profile after $D$'s deviation. We consider two sub-cases:

1. $I(\Sigma') < (2B + 2)\exp(-2C^2\alpha_C T)$, and
2. $I(\Sigma') \geq (2B + 2)\exp(-2C^2\alpha_C T)$.

In the first case, $I(\Sigma')$ is small, so the mechanism nearly always chooses the majority wish under $\Sigma'$. Therefore, the output of the mechanism does not change with high probability from profile $\Sigma^*$ to profile $\Sigma'$. In particular, we have $\lambda_H^{\mathbf{A}}(\Sigma') \approx \lambda_H^{\mathbf{A}}(\Sigma^*) \approx 1$ and $\lambda_L^{\mathbf{R}}(\Sigma') \approx \lambda_L^{\mathbf{R}}(\Sigma^*) \approx 1$. Thus, no agent can be much better off because all the agents have nearly the same utilities as before. This is formally proved in Claim B.3.

In the second case, $I(\Sigma')$ is not small, so the mechanism sometimes fails to choose the majority wish under $\Sigma'$. Here, the contingent agents do not fare better (Claim B.4) and thus no contingent agent can be in the deviating coalition. The technical key is then to show that $D$ cannot contain both a candidate-friendly and a candidate-unfriendly agent. Lemma B.1 shows that a significant increase in a candidate-friendly agent's (*ex-ante*) utility always results a decrease in a candidate-unfriendly agent's (*ex-ante*) utility, and vice versa. This is obvious if we are dealing with *ex-post* utilities, as a candidate-friendly agent and a candidate-unfriendly agent always want the opposite alternatives. However, this becomes much less obvious for *ex-ante* utilities.

Thus, any deviating coalition can only be comprised of either candidate-friendly agents or candidate-unfriendly agents. Finally, in Claim B.5, we show that a minority coalition comprised of only candidate-friendly agents cannot change the outcome by misreporting. This concludes the theorem.

### B.3.2 No Win-win Lemma

A key part of the proof is the following lemma which states that it is impossible that predetermined agents of different alternatives both gain from deviating from truthful. As a corollary, any deviating coalition can only contain predetermined agents of one type.

The proof of Lemma B.1 depends on i) truth-telling nearly always selecting the majority wish; and ii) the monotonicity of the *ex-post* utilities $v_t(\cdot, \mathbf{A})$ and $v_t(\cdot, \mathbf{R})$ in the first argument (here, we mean $v_t(L, \mathbf{A}) < v_t(H, \mathbf{A})$ and $v_t(L, \mathbf{R}) > v_t(H, \mathbf{R})$). In particular, Lemma B.1 does not hold if truth-telling is replaced by an arbitrary strategy profile.

**Lemma B.1.** *Suppose $\alpha_F < 0.5$ and $\alpha_U < 0.5$. Let $\Sigma^*$ be the truthful strategy profile and $\Sigma'$ be an arbitrary strategy profile. Let $t_1$ be an arbitrary candidate-friendly agent and $t_2$ be an arbitrary candidate-unfriendly agent. For any $\Delta \geq 2B\exp(-2c^2\alpha_C T)$, we have*

**(i)** *If $u_{t_1}(\Sigma') - u_{t_1}(\Sigma^*) > \Delta$, then $u_{t_2}(\Sigma') - u_{t_2}(\Sigma^*) < 0$.*

**(ii)** *If $u_{t_2}(\Sigma') - u_{t_2}(\Sigma^*) > \Delta$, then $u_{t_1}(\Sigma') - u_{t_1}(\Sigma^*) < 0$.*

*Proof.* We will only show (i), as the proof for (ii) is similar.

Since $v_t(L, \mathbf{A}) < v_t(H, \mathbf{A})$ and $v_t(L, \mathbf{R}) > v_t(H, \mathbf{R})$ for any agent $t$ (see (2)), we have $v_t(L, \mathbf{A}) - v_t(H, \mathbf{A}) < 0 < v_t(L, \mathbf{R}) - v_t(H, \mathbf{R})$, which further implies

$$v_t(L, \mathbf{A}) - v_t(L, \mathbf{R}) < v_t(H, \mathbf{A}) - v_t(H, \mathbf{R}).$$

Since a candidate-friendly agent always prefers $\mathbf{A}$ and a candidate-unfriendly agent always prefers $\mathbf{R}$, we have

$$\begin{aligned}
0 < v_{t_1}(L, \mathbf{A}) - v_{t_1}(L, \mathbf{R}) < v_{t_1}(H, \mathbf{A}) - v_{t_1}(H, \mathbf{R}), \text{ and}\\
v_{t_2}(L, \mathbf{A}) - v_{t_2}(L, \mathbf{R}) < v_{t_2}(H, \mathbf{A}) - v_{t_2}(H, \mathbf{R}) < 0.
\end{aligned} \tag{7}$$

By referring to (5), this intuitively says that $t_1$'s utility difference $u_{t_1}(\Sigma') - u_{t_1}(\Sigma^*)$ is more sensitive to $P_H \lambda_H^{\mathbf{A}}$ while $u_2$'s utility difference $u_{t_2}(\Sigma') - u_{t_2}(\Sigma^*)$ is more sensitive to $P_L \lambda_L^{\mathbf{A}}$.

Suppose $u_{t_1}(\Sigma') - u_{t_1}(\Sigma^*) > \Delta$ as it is assumed in (i). By (5), we have

$$\begin{aligned}
P_L(\lambda_L^{\mathbf{A}}(\Sigma') - \lambda_L^{\mathbf{A}}(\Sigma^*))(v_{t_1}(L, \mathbf{A}) - v_{t_1}(L, \mathbf{R}))\\
+P_H(\lambda_H^{\mathbf{A}}(\Sigma') - \lambda_H^{\mathbf{A}}(\Sigma^*))(v_{t_1}(H, \mathbf{A}) - v_{t_1}(H, \mathbf{R})) > \Delta.
\end{aligned} \tag{8}$$

We consider two cases: $\lambda_H^{\mathbf{A}}(\Sigma') \geq \lambda_H^{\mathbf{A}}(\Sigma^*)$ and $\lambda_H^{\mathbf{A}}(\Sigma') < \lambda_H^{\mathbf{A}}(\Sigma^*)$.

The intuitions for the remaining part of this proof is as follows. In the first case, the probability of outputting $\mathbf{A}$ (weakly) increases under world $H$. Since Theorem 3.2 tells us $\lambda_H^{\mathbf{A}}(\Sigma^*)$ is already close to

1, the utility gain for $t_1$ due to the increased probability of outputting $\mathbf{A}$ under world $H$ is insignificant, and we must still have $\lambda_L^{\mathbf{A}}(\Sigma') - \lambda_L^{\mathbf{A}}(\Sigma^*) > 0$ to ensure $u_{t_1}(\Sigma') - u_{t_1}(\Sigma^*) > \Delta$. However, since the probability of outputting $\mathbf{A}$ increases under both worlds, the utility for $t_2$ will decrease. In the second case, the probability of outputting $\mathbf{A}$ decreases under world $H$, which reduces the utility for $t_1$. To compensate this, the probability of outputting $\mathbf{A}$ under world $L$ must increase, in order to ensure $u_{t_1}(\Sigma') - u_{t_1}(\Sigma^*) > \Delta$. Moreover, we must have $P_L(\lambda_L^{\mathbf{A}}(\Sigma') - \lambda_L^{\mathbf{A}}(\Sigma^*)) > P_H(\lambda_H^{\mathbf{A}}(\Sigma^*) - \lambda_H^{\mathbf{A}}(\Sigma'))$ since the utility difference for agent $t_1$ is more sensitive to $P_H \lambda_H^{\mathbf{A}}$. However, since the utility difference for agent $t_2$ is more sensitive to $P_L \lambda_L^{\mathbf{A}}$, this will reduce the overall utility for $t_2$. These are formally proved below.

**Case 1:** $\lambda_H^{\mathbf{A}}(\Sigma') \geq \lambda_H^{\mathbf{A}}(\Sigma^*)$. By Theorem 3.2, we have $\lambda_H^{\mathbf{A}}(\Sigma^*) \geq 1 - 2\exp(-2c^2\alpha_C T)$, which implies $\lambda_H^{\mathbf{A}}(\Sigma') - \lambda_H^{\mathbf{A}}(\Sigma^*) \leq 2\exp(-2c^2\alpha_C T)$, which further implies

$$P_H(\lambda_H^{\mathbf{A}}(\Sigma') - \lambda_H^{\mathbf{A}}(\Sigma^*))(v_{t_1}(H, \mathbf{A}) - v_{t_1}(H, \mathbf{R})) \leq P_H \cdot 2\exp(-2c^2\alpha_C T) \cdot B < \Delta.$$

Putting this into (8), we have $P_L(\lambda_L^{\mathbf{A}}(\Sigma') - \lambda_L^{\mathbf{A}}(\Sigma^*))(v_{t_1}(L, \mathbf{A}) - v_{t_1}(L, \mathbf{R})) > 0$, which implies $\lambda_L^{\mathbf{A}}(\Sigma') > \lambda_L^{\mathbf{A}}(\Sigma^*)$. We then must have $u_{t_2}(\Sigma') - u_{t_2}(\Sigma^*) < 0$ since we have

$$\begin{aligned} u_{t_2}(\Sigma') - u_{t_2}(\Sigma^*) = & P_L(\lambda_L^{\mathbf{A}}(\Sigma') - \lambda_L^{\mathbf{A}}(\Sigma^*))(v_{t_2}(L, \mathbf{A}) - v_{t_2}(L, \mathbf{R})) \\ & + P_H(\lambda_H^{\mathbf{A}}(\Sigma') - \lambda_H^{\mathbf{A}}(\Sigma^*))(v_{t_2}(H, \mathbf{A}) - v_{t_2}(H, \mathbf{R})) \end{aligned}$$

by (5), $\lambda_H^{\mathbf{A}}(\Sigma') \geq \lambda_H^{\mathbf{A}}(\Sigma^*)$ (Case 1 assumption), $\lambda_L^{\mathbf{A}}(\Sigma') > \lambda_L^{\mathbf{A}}(\Sigma^*)$ (we have just shown), $v_{t_2}(L, \mathbf{A}) - v_{t_2}(L, \mathbf{R}) < 0$ and $v_{t_2}(H, \mathbf{A}) - v_{t_2}(H, \mathbf{R}) < 0$ (since $t_2$ is candidate-unfriendly).

**Case 2:** $\lambda_H^{\mathbf{A}}(\Sigma') < \lambda_H^{\mathbf{A}}(\Sigma^*)$. By (8) and $\Delta > 0$, we have

$$P_L(\lambda_L^{\mathbf{A}}(\Sigma') - \lambda_L^{\mathbf{A}}(\Sigma^*))(v_{t_1}(L, \mathbf{A}) - v_{t_1}(L, \mathbf{R})) > P_H(\lambda_H^{\mathbf{A}}(\Sigma^*) - \lambda_H^{\mathbf{A}}(\Sigma'))(v_{t_1}(H, \mathbf{A}) - v_{t_1}(H, \mathbf{R})),$$

which, by the first inequality in (7), further implies

$$P_L(\lambda_L^{\mathbf{A}}(\Sigma') - \lambda_L^{\mathbf{A}}(\Sigma^*)) > P_H(\lambda_H^{\mathbf{A}}(\Sigma^*) - \lambda_H^{\mathbf{A}}(\Sigma')) > 0.$$

By the second inequality in (7), this implies

$$P_L(\lambda_L^{\mathbf{A}}(\Sigma') - \lambda_L^{\mathbf{A}}(\Sigma^*))(v_{t_2}(L, \mathbf{A}) - v_{t_2}(L, \mathbf{R})) < P_H(\lambda_H^{\mathbf{A}}(\Sigma^*) - \lambda_H^{\mathbf{A}}(\Sigma'))(v_{t_2}(H, \mathbf{A}) - v_{t_2}(H, \mathbf{R})),$$

which further implies

$$\begin{aligned} u_{t_2}(\Sigma') - u_{t_2}(\Sigma^*) = & P_L(\lambda_L^{\mathbf{A}}(\Sigma') - \lambda_L^{\mathbf{A}}(\Sigma^*))(v_{t_2}(L, \mathbf{A}) - v_{t_2}(L, \mathbf{R})) \\ & + P_H(\lambda_H^{\mathbf{A}}(\Sigma') - \lambda_H^{\mathbf{A}}(\Sigma^*))(v_{t_2}(H, \mathbf{A}) - v_{t_2}(H, \mathbf{R})) < 0. \end{aligned}$$

The lemma concludes. $\qquad\square$

**Corollary B.2.** *Suppose $\alpha_F < 0.5$ and $\alpha_U < 0.5$. The set of deviating agents $D$ cannot contain both candidate friendly and candidate unfriendly agents.*

*Proof.* By 3 in Definition 2.1, there must be an agent $t$ such that $u_t(\Sigma') - u_t(\Sigma^*) > \varepsilon \geq 2B\exp(-2c^2\alpha_C T)$. Assume this agent is candidate-friendly. Then by Lemma B.1, for any candidate-unfriendly agent $t'$, we have $u_{t'}(\Sigma') - u_{t'}(\Sigma^*) < 0$. Thus no candidate-unfriendly agent can be in the deviating coalition.

An analogous argument works if the benefiting agent is candidate-unfriendly. $\qquad\square$

### B.3.3 Proof of Theorem 3.3

Now we are ready to prove Theorem 3.3.

Suppose this is not the case. There exists a set of deviating agents $D$ that can deviate from the truthful strategy such that all of them receive utilities that are at least their original utilities and some of them receive utilities that are $\varepsilon$ strictly higher than their original utilities. Let $\Sigma'$ be the strategy profile after agents in $D$ deviate.

We discuss three different cases: 1) $\alpha_F > 0.5$, 2) $\alpha_U > 0.5$ and 3) $\alpha_F < 0.5$ and $\alpha_U < 0.5$. Notice that $n$ being odd implies neither $\alpha_F$ nor $\alpha_U$ can be exactly 0.5.

**Case 1:** $\alpha_F > 0.5$. If all agents report truthfully, $\mathbf{A}$ will be announced with probability 1 according to Step 5 of the mechanism. That is, $\lambda_L^{\mathbf{A}}(\Sigma^*) = \lambda_H^{\mathbf{A}}(\Sigma^*) = 1$. By 3 of Definition 2.1, there exists $t \in D$ such that $u_t(\Sigma') > u_t(\Sigma^*) + \varepsilon$. Since $\lambda_L^{\mathbf{A}}(\Sigma')$ and $\lambda_H^{\mathbf{A}}(\Sigma')$ completely determine each agent's utility, we must have either $\lambda_L^{\mathbf{A}}(\Sigma') \neq \lambda_L^{\mathbf{A}}(\Sigma^*)$ or $\lambda_H^{\mathbf{A}}(\Sigma') \neq \lambda_H^{\mathbf{A}}(\Sigma^*)$. This means either $\lambda_L^{\mathbf{A}}(\Sigma') < 1$ or $\lambda_H^{\mathbf{A}}(\Sigma') < 1$.

Since a candidate-friendly agent's utility is maximized when both $\lambda_L^{\mathbf{A}}$ and $\lambda_H^{\mathbf{A}}$ are 1, a candidate-friendly agent's utility will decrease if the strategy profile is switched from $\Sigma^*$ to $\Sigma'$. By 2 of Definition 2.1, $D$ does not contain any candidate-friendly agent. However, if this is the case, there are still more than half of the agents that will report $F$ (as $\alpha_F > 0.5$), and $\mathbf{A}$ will always be announced by Step 5 of the mechanism. We conclude that $\lambda_L^{\mathbf{A}}(\Sigma') = \lambda_H^{\mathbf{A}}(\Sigma') = 1$, which contradicts what we have concluded in the previous paragraph.

**Case 2:** $\alpha_U > 0.5$. The analysis is similar to the previous case.

**Case 3:** $\alpha_F < 0.5$ **and** $\alpha_U < 0.5$. We consider two sub-cases: $I(\Sigma') < (2B + 2)\exp(-2c^2\alpha_C T)$ and $I(\Sigma') \geq (2B + 2)\exp(-2c^2\alpha_C T)$

Firstly, we consider $I(\Sigma') < (2B + 2)\exp(-2c^2\alpha_C T)$.

**Claim B.3.** *If $\alpha_F < 0.5$, $\alpha_U < 0.5$ and $I(\Sigma') < (2B+2)\exp(-2c^2\alpha_C T)$, then $u_t(\Sigma') - u_t(\Sigma) < \varepsilon$ for every agent $t$.*

The ideas behind this proof is that the outcome for $\Sigma'$ is too close to the outcome of the truthful strategy profile $\Sigma^*$, so no agent can get significantly more benefit.

*Proof.* The proof of this claim shows that none of the three types of agents can benefit by $\varepsilon$ because nothing is substantially different from when agents play truthfully.

By the inequality $I(\Sigma') = P_L \lambda_L^{\mathbf{A}}(\Sigma') + P_H \lambda_H^{\mathbf{R}}(\Sigma') < (2B + 2)\exp(-2c^2\alpha_C T)$, we have

$$0 \leq \lambda_H^{\mathbf{R}}(\Sigma') \leq \frac{(2B + 2)\exp(-2c^2\alpha_C T)}{P_H}$$

and

$$0 \leq \lambda_L^{\mathbf{A}}(\Sigma') \leq \frac{(2B + 2)\exp(-2c^2\alpha_C T)}{P_L}.$$

Since $\lambda_L^{\mathbf{A}}(\Sigma^*) \leq 2\exp(-2c^2\alpha_C T)$ and $\lambda_H^{\mathbf{R}}(\Sigma^*) \leq 2\exp(-2c^2\alpha_C T)$ by Theorem 3.2, we have

$$\lambda_L^{\mathbf{A}}(\Sigma^*) - \lambda_L^{\mathbf{A}}(\Sigma') \leq \lambda_L^{\mathbf{A}}(\Sigma^*) \leq 2\exp(-2c^2\alpha_C T),$$

$$\lambda_L^{\mathbf{A}}(\Sigma') - \lambda_L^{\mathbf{A}}(\Sigma^*) \leq \lambda_L^{\mathbf{A}}(\Sigma') \leq \frac{(2B + 2)\exp(-2c^2\alpha_C T)}{P_L},$$

$$\lambda_H^{\mathbf{A}}(\Sigma^*) - \lambda_H^{\mathbf{A}}(\Sigma') \leq 1 - \left(1 - \lambda_H^{\mathbf{R}}(\Sigma')\right) \leq \frac{(2B + 2)\exp(-2c^2\alpha_C T)}{P_H},$$

and

$$\lambda_H^{\mathbf{A}}(\Sigma') - \lambda_H^{\mathbf{A}}(\Sigma^*) \leq 1 - \left(1 - \lambda_H^{\mathbf{R}}(\Sigma^*)\right) \leq 2\exp(-2c^2\alpha_C T).$$

Now, we substitute the four inequalities into the following equation implied by (5):

$$\begin{aligned}u_t(\Sigma') - u_t(\Sigma^*) &= P_L(\lambda_L^{\mathbf{A}}(\Sigma') - \lambda_L^{\mathbf{A}}(\Sigma^*))(v_t(L, \mathbf{A}) - v_t(L, \mathbf{R})) \\ &+ P_H(\lambda_H^{\mathbf{A}}(\Sigma') - \lambda_H^{\mathbf{A}}(\Sigma^*))(v_t(H, \mathbf{A}) - v_t(H, \mathbf{R})).\end{aligned}$$

For any candidate-friendly agent, we have $v_t(L, \mathbf{A}) > v_t(L, \mathbf{R})$ and $v_t(H, \mathbf{A}) > v_t(H, \mathbf{R})$, which yields

$$\begin{aligned}&u_t(\Sigma') - u_t(\Sigma^*) \\ =&P_L(\lambda_L^{\mathbf{A}}(\Sigma') - \lambda_L^{\mathbf{A}}(\Sigma^*))(v_t(L, \mathbf{A}) - v_t(L, \mathbf{R})) + P_H(\lambda_H^{\mathbf{A}}(\Sigma') - \lambda_H^{\mathbf{A}}(\Sigma^*))(v_t(H, \mathbf{A}) - v_t(H, \mathbf{R})) \\ \leq&P_L \cdot \frac{(2B + 2)\exp(-2c^2\alpha_C T)}{P_L} \cdot B + P_H \cdot 2\exp(-2c^2\alpha_C T)) \cdot B \\ <&(2B^2 + 4B)\exp(-2c^2\alpha_C T) = \varepsilon.\end{aligned}$$

For any contingent agent, we have $v_t(L, \mathbf{R}) > v_t(L, \mathbf{A})$ and $v_t(H, \mathbf{A}) > v_t(H, \mathbf{R})$, which yields

$$
\begin{aligned}
&u_t(\Sigma') - u_t(\Sigma^*) \\
=&P_L(\lambda_L^{\mathbf{A}}(\Sigma^*) - \lambda_L^{\mathbf{A}}(\Sigma'))(v_t(L, \mathbf{R}) - v_t(L, \mathbf{A})) + P_H(\lambda_H^{\mathbf{A}}(\Sigma') - \lambda_H^{\mathbf{A}}(\Sigma^*))(v_t(H, \mathbf{A}) - v_t(H, \mathbf{R})) \\
\leq&P_L \cdot 2\exp(-2c^2\alpha_C T) \cdot B + P_H \cdot 2\exp(-2c^2\alpha_C T) \cdot B \\
=&2B\exp(-2c^2\alpha_C T) < \varepsilon.
\end{aligned}
$$

For any candidate-unfriendly agent, we have $v_t(L, \mathbf{R}) > v_t(L, \mathbf{A})$ and $v_t(H, \mathbf{R}) > v_t(H, \mathbf{A})$, which yields

$$
\begin{aligned}
&u_t(\Sigma') - u_t(\Sigma^*) \\
=&P_L(\lambda_L^{\mathbf{A}}(\Sigma^*) - \lambda_L^{\mathbf{A}}(\Sigma'))(v_t(L, \mathbf{R}) - v_t(L, \mathbf{A})) + P_H(\lambda_H^{\mathbf{A}}(\Sigma^*) - \lambda_H^{\mathbf{A}}(\Sigma'))(v_t(H, \mathbf{R}) - v_t(H, \mathbf{A})) \\
\leq&P_L \cdot 2\exp(-2c^2\alpha_C T) \cdot B + P_H \cdot \frac{(2B+2)\exp(-2c^2\alpha_C T)}{P_H} \cdot B \\
<&(2B^2 + 4B)\exp(-2c^2\alpha_C T) = \varepsilon.
\end{aligned}
$$

We conclude none of the agents has a utility gain of at least $\varepsilon$, which contradict 3 of Definition 2.1. $\quad\square$

Claim B.3 implies that, in the first case $I(\Sigma') < (2B+2)\exp(-2c^2\alpha_C T)$, there does not exist a deviating coalition $D$ where an agent in $D$ can receive an utility gain of at least $\varepsilon$, which contradicts to our assumption about $D$ at the beginning.

Next, we consider the second case $I(\Sigma') \geq (2B+2)\exp(-2c^2\alpha_C T)$. This case is more complicated. We first show that no contingent agent in $\Sigma'$ can do as well as in the truthful profile $\Sigma^*$.

**Claim B.4.** *If* $\alpha_F < 0.5$, $\alpha_U < 0.5$ *and* $I(\Sigma') \geq (2B+2)\exp(-2c^2\alpha_C T)$, *then* $u_t(\Sigma') - u_t(\Sigma^*) < 0$ *for every contingent agent* $t$.

The ideas behind this proof is that those contingent agents already receive almost optimal utilities in $\Sigma^*$; therefore, if the error rate of the strategy $\Sigma'$ is high enough, the utilities of the contingent agents will decrease.

*Proof.* By Theorem 3.2, we have

$$
\begin{aligned}
P_L\lambda_L^{\mathbf{R}}(\Sigma^*) + P_H\lambda_H^{\mathbf{A}}(\Sigma^*) &\geq P_L(1 - 2\exp(-2c^2\alpha_C T)) + P_H(1 - 2\exp(-2c^2\alpha_C T)) \\
&= 1 - 2\exp(-2c^2\alpha_C T). \tag{9}
\end{aligned}
$$

Therefore

$$
\begin{aligned}
P_L\lambda_L^{\mathbf{R}}(\Sigma^*) + P_H\lambda_H^{\mathbf{A}}(\Sigma^*) &\geq 1 - 2\exp(-2c^2\alpha_C T) \\
&= 2B\exp(-2c^2\alpha_C T) + 1 - (2B+2)\exp(-2c^2\alpha_C T) \\
&\geq 2B\exp(-2c^2\alpha_C T) + 1 - I(\Sigma') \\
&= P_L\lambda_L^{\mathbf{R}}(\Sigma') + P_H\lambda_H^{\mathbf{A}}(\Sigma') + 2B\exp(-2c^2\alpha_C T) \tag{10}
\end{aligned}
$$

By rewriting (4) as (noticing $\lambda_L^{\mathbf{A}}(\Sigma) + \lambda_H^{\mathbf{A}}(\Sigma) = 1$)

$$
\begin{aligned}
u_t(\Sigma) = &P_L v_t(L, \mathbf{A}) + P_H v_t(H, \mathbf{R}) + P_L\lambda_L^{\mathbf{R}}(\Sigma)(v_t(L, \mathbf{R}) - v_t(L, \mathbf{A})) \\
&+ P_H\lambda_H^{\mathbf{A}}(\Sigma)(v_t(H, \mathbf{A}) - v_t(H, \mathbf{R})),
\end{aligned}
$$

we have

$$
\begin{aligned}
u_t(\Sigma') - u_t(\Sigma^*) = &P_L(v_t(L, \mathbf{R}) - v_t(L, \mathbf{A}))(\lambda_L^{\mathbf{R}}(\Sigma') - \lambda_L^{\mathbf{R}}(\Sigma^*)) \\
&+ P_H(v_t(H, \mathbf{A}) - v_t(H, \mathbf{R}))(\lambda_H^{\mathbf{A}}(\Sigma') - \lambda_H^{\mathbf{A}}(\Sigma^*))
\end{aligned} \tag{11}
$$

We will show $u_t(\Sigma') - u_t(\Sigma^*) < 0$ for an arbitrary contingent agent $t$. Recall that $v_t(L, \mathbf{R}) - v_t(L, \mathbf{A}) > 0$ and $v_t(H, \mathbf{A}) - v_t(H, \mathbf{R}) > 0$. We consider three cases:

- If $\lambda_H^{\mathbf{A}}(\Sigma') \leq \lambda_H^{\mathbf{A}}(\Sigma^*)$ and $\lambda_L^{\mathbf{R}}(\Sigma') \leq \lambda_L^{\mathbf{R}}(\Sigma^*)$, then one of these two inequalities must be strict by (10). Equation (11) then implies $u_t(\Sigma') - u_t(\Sigma^*) < 0$.

- If $\lambda_H^{\mathbf{A}}(\Sigma') > \lambda_H^{\mathbf{A}}(\Sigma^*)$, then we have $P_L \lambda_L^{\mathbf{R}}(\Sigma^*) - P_L \lambda_L^{\mathbf{R}}(\Sigma') > 2B \exp(-2c^2 \alpha_C T)$ by (10). Since $\lambda_H^{\mathbf{A}}(\Sigma') \leq 1$ and $\lambda_H^{\mathbf{A}}(\Sigma^*) \geq 1 - 2\exp(-2c^2 \alpha_C T)$, we have $\lambda_H^{\mathbf{A}}(\Sigma') - \lambda_H^{\mathbf{A}}(\Sigma^*) \leq 2\exp(-2c^2 \alpha_C T)$. We also have $v_t(H, \mathbf{A}) - v_t(H, \mathbf{R}) \leq B$ and $v_t(L, \mathbf{R}) - v_t(L, \mathbf{A}) \geq 1$ (recall that $v_t(L, \mathbf{A})$, $v_t(L, \mathbf{R})$, $v_t(H, \mathbf{R})$ and $v_t(H, \mathbf{A})$ are integers bounded by $B$). Putting those into (11), we have

$$u_t(\Sigma') - u_t(\Sigma^*) < 1 \cdot (-2B \exp(-2c^2 \alpha_C T)) + P_H \cdot B \cdot 2\exp(-2c^2 \alpha_C T) < 0.$$

- If $\lambda_L^{\mathbf{R}}(\Sigma') > \lambda_L^{\mathbf{R}}(\Sigma^*)$, then we have $P_H \lambda_H^{\mathbf{A}}(\Sigma^*) - P_H \lambda_H^{\mathbf{A}}(\Sigma') > 2B \exp(-2c^2 \alpha_C T)$ by (10). Similar to the second case, we have $\lambda_L^{\mathbf{R}}(\Sigma') - \lambda_L^{\mathbf{R}}(\Sigma^*) \leq 2\exp(-2c^2 \alpha_C T)$, $v_t(H, \mathbf{A}) - v_t(H, \mathbf{R}) \geq 1$ and $v_t(L, \mathbf{R}) - v_t(L, \mathbf{A}) \leq B$. Putting those into (11), we have

$$u_t(\Sigma') - u_t(\Sigma^*) < P_L \cdot B \cdot 2\exp(-2c^2 \alpha_C T) + 1 \cdot (-2B \exp(-2c^2 \alpha_C T)) < 0.$$

Putting these three cases together, we have $u_t(\Sigma') - u_t(\Sigma^*) < 0$ for an arbitrary contingent agent $t$. $\qquad\square$

Therefore, $D$ cannot contain any contingent agents by 2 of Definition 2.1. Corollary B.2 says that $D$ cannot simultaneously contain an candidate-friendly agent and a candidate-unfriendly agent. Thus any $D$ must contain either only candidate-friendly agents or only candidate-unfriendly agents.

The following claim states that neither type of predetermined agents alone are powerful enough to change the outcome to their favor. This concludes the proof as we have shown there is no deviating coalition.

**Claim B.5.** *Suppose $\alpha_F < 0.5$ and $\alpha_U < 0.5$. If $D$ contains only candidate-friendly agents, then $\lambda_L^A(\Sigma') \leq \lambda_L^A(\Sigma^*)$ and $\lambda_H^A(\Sigma') \leq \lambda_H^A(\Sigma^*)$. If $D$ contains only candidate-unfriendly agents, then $\lambda_L^R(\Sigma') \leq \lambda_L^R(\Sigma^*)$ and $\lambda_H^R(\Sigma') \leq \lambda_H^R(\Sigma^*)$.*

*Proof.* Consider candidate-friendly agents without loss of generality. Since contingent agents and candidate-unfriendly agents, which constitute more than half of the population, are truth-telling, those candidate-friendly agents cannot make the mechanism announce $\mathbf{A}$ at Step 5, since they cannot make more than half of agents report type $F$. To maximize the probability that the mechanism announce $\mathbf{A}$ at Step 6, those candidate-friendly agents would like to maximize the fraction of agents reporting signal $h$ and minimize the median $\bar{\delta}$. However, the mechanism's conversion of signals (Step 2) and predictions (Step 3) already does these for candidate-friendly agents. $\qquad\square$

Thus, we have proved that, for the second case $I(\Sigma') \geq (2B + 2)\exp(-2c^2 \alpha_C T)$, no such deviating set $D$ exists, which contradicts to our assumption for the existence of $D$ at the beginning.

# C An Alternative Mechanism

As we have remarked right below Mechanism 1, we present an alternative mechanism that achieves the same theoretical properties. The mechanism is shown in Mechanism 2.

---

**Mechanism 2** The Wisdom-of-the-Crowd-Voting Mechanism (an alternative)

---

1: Each agent $t$ reports to the mechanism the signal (s)he receives (either $\ell$ or $h$), denoted by $\bar{s}_i \in \{\ell, h\}$, his/her type ($F$, $U$ or $C$), his/her posterior belief of the fraction of agents who will report signal $h$, denoted by $\bar{\delta}_t$.
2: If agent $t$ reports type $F$, his reported signal will be automatically treated as $\bar{s}_t = h$; if agent $t$ reports type $U$, his reported signal will be automatically treated as $\bar{s}_t = \ell$. *The prediction $\bar{\delta}_t$ in the previous step should be made with this treatment being considered, and the mechanism makes this clear to the agents.*
3: Compute the *median* of the reported $\bar{\delta}_t$, denoted by $\bar{\delta}$.
4: If more than half of the agents report type $F$, announce $\mathbf{A}$ being the winning alternative; if more than half of the agents reports type $U$, announce $\mathbf{R}$ being the winning alternative.
5: If the number of agents reporting $\bar{s}_t = h$ is more than the median $\bar{\delta}$, announce $\mathbf{A}$ being the winning alternative; otherwise, announce $\mathbf{R}$ being the winning alternative.

---

Correspondingly, the questionnaire becomes the followings.

1. Choose one of the followings:

    (a) I definitely want to accept this candidate.
    (b) I definitely want to reject this candidate.
    (c) After talking to the candidate, I am more inclined to accept him/her than before.
    (d) After talking to the candidate, I am more inclined to reject him/her than before.

2. What percentage of the faculty members do you believe will choose (a) or (c) in the first question?

Theorem 3.2 still holds for Mechanism 2. If $\alpha_F > 0.5$ or $\alpha_U > 0.5$, the mechanism outputs the majority wish (**A** or **R** respectively) at Step 4 with probability 1 as before. If $\alpha_F, \alpha_U < 0.5$, we still have $\bar{\delta} \in [\alpha_C T_{h\ell} + \alpha_F, \alpha_C T_{hh} + \alpha_F]$. This is actually easier to see: agents' predictions are now either $\alpha_C T_{h\ell} + \alpha_F$ or $\alpha_C T_{hh} + \alpha_F$. The remaining part of the proof is the same as before.

Theorem 3.3 still holds for Mechanism 2. In fact, all parts of the proof are the same as before, except for Claim B.5 where we can only prove a weaker statement, which is, nevertheless, sufficient to show Theorem 3.3.

**Claim C.1.** *Suppose $\alpha_F < 0.5$ and $\alpha_U < 0.5$. If $D$ contains only candidate-friendly agents, then $\lambda_L^{\textbf{A}}(\Sigma') \leq \lambda_L^{\textbf{A}}(\Sigma^*) + 2\exp(-2c^2\alpha_C T)$ and $\lambda_H^{\textbf{A}}(\Sigma') \leq \lambda_H^{\textbf{A}}(\Sigma^*) + 2\exp(-2c^2\alpha_C T)$. If $D$ contains only candidate-unfriendly agents, then $\lambda_L^{\textbf{R}}(\Sigma') \leq \lambda_L^{\textbf{R}}(\Sigma^*) + 2\exp(-2c^2\alpha_C T)$ and $\lambda_H^{\textbf{R}}(\Sigma') \leq \lambda_H^{\textbf{R}}(\Sigma^*) + 2\exp(-2c^2\alpha_C T)$.*

*Proof.* We focus on the case that $D$ contains only candidate-friendly agents. The candidate-unfriendly case is analogous.

First of all, since there are strictly less than half of the agents reporting type $F$ (those type $U$ and type $C$ agents, which contribute more than half of the population, report their types truthfully), those candidate-friendly agents in $D$ cannot make the mechanism output **A** at Step 5 of the mechanism. Therefore, they can only attempt to make the mechanism output **A** at Step 6 with a higher probability.

Suppose the actual world is $H$. We need to prove $\lambda_H^{\textbf{A}}(\Sigma') \leq \lambda_H^{\textbf{A}}(\Sigma^*) + 2\exp(-2c^2\alpha_C T)$. This is trivial: Theorem 3.2 implies $\lambda_H^{\textbf{A}}(\Sigma^*) \geq 1 - 2\exp(-2c^2\alpha_C T)$, so the right-hand side of the inequality is at least 1, making the inequality always hold.

Suppose the actual world is $L$. We need to prove $\lambda_L^{\textbf{A}}(\Sigma') \leq \lambda_L^{\textbf{A}}(\Sigma^*) + 2\exp(-2c^2\alpha_C T)$. It suffices to show $\lambda_L^{\textbf{A}}(\Sigma') \leq 2\exp(-2c^2\alpha_C T)$, which is equivalent to

$$\lambda_L^{\textbf{R}}(\Sigma') \geq 1 - 2\exp(-2c^2\alpha_C T). \tag{12}$$

Supposing $\Sigma'$ is played, we will prove the following two observations:

1. With probability at least $1 - 2\exp(-2c^2\alpha_C T)$, the fraction of agents reporting signal $h$ is at most $\alpha_C(P_{hL} + c) + \alpha_F$;

2. The median of the prediction $\bar{\delta}$ falls into the interval $[\alpha_C T_{h\ell} + \alpha_F, \alpha_C T_{hh} + \alpha_F]$ (with probability 1).

To show the first observation, all the candidate-unfriendly agents will report signal $\ell$ (after the conversion in Step 2). For the contingent agents, each of them receives signal $h$ with probability $P_{hL}$. By a Chernoff bound, with probability at least $1 - 2\exp(-2c^2\alpha_C T)$, the fraction of the contingent agents receiving $h$ is at most $P_{hL} + c$. Even if all the candidate-friendly agents report $h$, the overall fraction of agents reporting $h$ is at most $\alpha_C(P_{hL} + c) + \alpha_F$ with probability at least $1 - 2\exp(-2c^2\alpha_C T)$.

To show the second observation, the prediction $\bar{\delta}_t$ that a truthful agent will report is either $\alpha_C T_{h\ell} + \alpha_F$ (if (s)he receive signal $\ell$) or $\alpha_C T_{hh} + \alpha_F$ (if (s)he receive signal $h$). Since there are more than half of truth-telling agents, the median $\bar{\delta}$ is always within the interval $[\alpha_C T_{h\ell} + \alpha_F, \alpha_C T_{hh} + \alpha_F]$.

Finally, by noticing $\alpha_C(P_{hL} + c) + \alpha_F < \alpha_C T_{h\ell} + \alpha_F$ (implied by Theorem 3.1 and (6)), the two observations imply that the fraction of agents reporting signal $h$ is strictly less than $\bar{\delta}$ with probability at least $1 - 2\exp(-2c^2\alpha_C T)$, which implies (12) by our design in Step 6 of the mechanism. $\square$

To conclude Theorem 3.3, for each agent $t$ in $D$ that contains only candidate-friendly agents, we have

$$
\begin{aligned}
&u_t(\Sigma') - u_t(\Sigma^*) \\
=&P_L(\lambda_L^{\mathbf{A}}(\Sigma') - \lambda_L^{\mathbf{A}}(\Sigma^*))(v_t(L, \mathbf{A}) - v_t(L, \mathbf{R})) + P_H(\lambda_H^{\mathbf{A}}(\Sigma') - \lambda_H^{\mathbf{A}}(\Sigma^*))(v_t(H, \mathbf{A}) - v_t(H, \mathbf{R})) \\
&\hspace{10cm} \text{(by (5))} \\
\leq&P_L \cdot 2\exp(-2c^2\alpha_C T) \cdot B + P_H \cdot 2\exp(-2c^2\alpha_C T) \cdot B \hspace{2cm} \text{(by Claim C.1)} \\
=&2B\exp(-2c^2\alpha_C T) < \varepsilon.
\end{aligned}
$$

Thus, no agent in $D$ satisfies 3 in Definition 2.1. A similar analysis holds for the case where $D$ contains only candidate-unfriendly agents.

### C.1   Comparison of the Two Mechanisms

Both Mechanism 1 and Mechanism 2 achieve the same set of theoretical properties.

The advantage for Mechanism 1 is that it is "slightly more truthful" in the sense that Claim B.5 is stronger than Claim C.1. In fact, under Mechanism 1, we have seen that truth-telling is a dominant strategy for both candidate-friendly and candidate-unfriendly agents, while this nice property is lost in Mechanism 2. Under Mechanism 2, the dominant strategy for a candidate-friendly agent (candidate-unfriendly agent resp.) is to report prediction 0 (1 resp.), which is no longer a truthful strategy. Nevertheless, we have seen that the truthful strategy is good enough so that a deviation to the dominant strategy does not provide a utility gain of at least $\varepsilon$.

Mechanism 2 wins by a little bit for its simplicity and symmetry. It is easier to explain Mechanism 2 to the users in practice. Notice that Mechanism 1 essentially converts the prediction reported from each candidate-friendly agent (candidate-unfriendly agent resp.) to 0 (1 resp.). Converting the predictions may seem to be less natural than converting the signals for users. Especially, for those users who are not familiar to the idea of "surprisingly popular", they may not be able to realize that converting their predictions to the opposite extreme is helpful for them, and they may be more skeptical of Mechanism 1 due to this. In addition, Mechanism 2 treats the reported predictions symmetrically, which may be more acceptable to the users in practical implementations.

Another related question is how exactly to phrase the ballot in practice. Here, we mimic the questions of Prelec et al. [2017] and ask for a forecast. However, it may be preferable in practice to ask, as in Mechanism 3, for a fractional threshold of (a) and (c) responses above which the agents would prefer to accept. While the outcomes would be mathematically equivalent, one or the other or a third alternative might work better in practice. This is, however, beyond the scope of this paper.

## D   Unknown/Partially Known Distribution of Agent Types

Before generalizing our result to the setting with general non-binary signals and states, we take a detour and discuss the necessity of the assumption that the distribution of agent types is common knowledge. As we mentioned in Section 2, we assume the distribution of agent types, $\alpha_F, \alpha_U, \alpha_C$, is a common knowledge among the agents. This assumption is natural by its own in many scenarios including our candidate hiring example (if a theory candidate is applying at a computer science department, those theory faculty members are more inclined to accept the candidate than the faculty members in AI, software, hardware; moreover, the numbers of the theory, AI, software, hardware faculty members are public information). In this section, we will see that this assumption is also necessary for the existence of a mechanism that satisfies Theorem 3.2 and Theorem 3.3.

Before describing our impossibility result, we first formally define the model with unknown agent types. Let $\Delta_3 = \{(x_1, x_2, x_3) \mid \forall i : x_i \in [0, 1]; x_1 + x_2 + x_3 = 1\}$. The distribution of types, $(\alpha_F, \alpha_U, \alpha_C)$, is then an element of $\Delta_3$. To model an unknown/partially known distribution of agent types, let $\mathcal{D}_{\Delta_3}$ be a distribution over $\Delta_3$ where each agent believes the distribution of the agent types, $(\alpha_F, \alpha_U, \alpha_C)$, is drawn from $\mathcal{D}_{\Delta_3}$.

Note that while the fraction of types is not known, the prior over the worlds, $P_L, P_H$, and the signal structures conditioned on types, $P_{\ell L}, P_{\ell H}, P_{hL}, P_{hH}$ are still common knowledge.

Next, we describe a natural property that is shared by most social choice mechanism, including the one in this paper.

**Definition D.1.** A mechanism is *anonymous* if it always outputs the same alternative for any two collections of reports $\mathbf{r}^{(1)} = (r_1^{(1)}, \ldots, r_T^{(1)}) \in \mathcal{R}^T, \mathbf{r}^{(2)} = (r_1^{(2)}, \ldots, r_T^{(2)}) \in \mathcal{R}^T$ such that $\mathbf{r}^{(1)}$ is a permutation of $\mathbf{r}^{(2)}$.

In other words, an anonymous mechanism cannot decide the output alternative based on agents' identities.

We have the following strong impossibility result.

**Theorem D.2.** *Under the setting with an unknown distribution of agent types, there exists a constant $\tau > 0$ such that no anonymous mechanism always outputs the alternative favored by more than half of the agents with probability more than $1 - \tau$ in any $\tau$-strong symmetric Bayes Nash equilibrium.*

Since the truthful strategy profile is symmetric, we have the following corollary about the impossibility of a truthful mechanism.

**Corollary D.3.** *Under the setting with unknown distribution of agent types defined above, there exists a constant $\tau > 0$ such that no anonymous mechanism satisfies both of the followings:*

- *the mechanism outputs the alternative favored by more than half of the agents with probability more than $1 - \tau$;*

- *under the mechanism, the truthful strategy profile is a $\tau$-strong Bayes Nash equilibrium.*

## D.1 Proof of Theorem D.2

Consider an anonymous mechanism and an arbitrary symmetric strategy profile $\Sigma$. Let $\beta_{F,\ell}$ be the fraction of candidate-friendly agents that receive signal $\ell$. Let $\beta_{F,h}, \beta_{U,\ell}, \beta_{U,h}, \beta_{C,\ell}$ and $\beta_{C,h}$ have similar meanings. The mechanism can only see how many different reports there are, and how many agents report each of them; in particular, the mechanism cannot see who reports which. The following proposition follows immediately from the above remarks.

**Proposition D.4.** *Fix a symmetric strategy profile $\Sigma$. If a mechanism is anonymous, then the values $\{\alpha_F, \alpha_U, \alpha_C\} \cup \{\beta_{F,\ell}, \beta_{F,h}, \beta_{U,\ell}, \beta_{U,h}, \beta_{C,\ell}, \beta_{C,h}\}$ completely determine the output of the mechanism.*

We also need the following technical lemma.

**Lemma D.5.** *The total variation distance between the two binomial distributions $\mathrm{Bin}(T, 1/6)$ and $\mathrm{Bin}(T/3, 1/2)$ is less than $0.123$ for sufficiently large $T$.*

*Proof.* By Central Limit Theorem, the total variation distance between $\mathrm{Bin}(T, 1/6)$ and $\mathrm{Bin}(T/3, 1/2)$ is at most the total variation distance between $\mathcal{N}(T/6, 5T/36)$ and $\mathcal{N}(T/6, T/12)$ plus $o(1)$, which, by shifting the mean of the Gaussian distribution, is the total variation distance between $\mathcal{N}(0, 5T/36)$ and $\mathcal{N}(0, T/12)$ plus $o(1)$.

Let $f(x)$ and $g(x)$ be the probability density function for $\mathcal{N}(0, 5T/36)$ and $\mathcal{N}(0, T/12)$ respectively. To calculate the total variation distance, firstly, straightforward calculations reveal that $f(x) < g(x)$ on $\left(-\sqrt{\frac{5}{24} \ln \frac{5}{3} T}, \sqrt{\frac{5}{24} \ln \frac{5}{3} T}\right)$ and $f(x) > g(x)$ on $\left(-\infty, -\sqrt{\frac{5}{24} \ln \frac{5}{3} T}\right) \cup \left(\sqrt{\frac{5}{24} \ln \frac{5}{3} T}, \infty\right)$. Therefore, the total variation distance between $\mathcal{N}(0, 5T/36)$ and $\mathcal{N}(0, T/12)$ is

$$
\int_{-\sqrt{\frac{5}{24} \ln \frac{5}{3} T}}^{\sqrt{\frac{5}{24} \ln \frac{5}{3} T}} g(x) - f(x) dx = \int_{-\sqrt{\frac{5}{24} \ln \frac{5}{3} T}}^{\sqrt{\frac{5}{24} \ln \frac{5}{3} T}} \frac{1}{\sqrt{2\pi \frac{T}{12}}} e^{-\frac{1}{2} \frac{x^2}{T/12}} - \frac{1}{\sqrt{2\pi \frac{5T}{36}}} e^{-\frac{1}{2} \frac{x^2}{5T/36}} dx
$$

$$
= \int_{-\sqrt{\frac{5}{24} \ln \frac{5}{3}}}^{\sqrt{\frac{5}{24} \ln \frac{5}{3}}} \frac{1}{\sqrt{2\pi \frac{1}{12}}} e^{-\frac{1}{2} \frac{y^2}{1/12}} - \frac{1}{\sqrt{2\pi \frac{5}{36}}} e^{-\frac{1}{2} \frac{y^2}{5/36}} dy
$$

$$
\text{(where } y = x/\sqrt{T})
$$

$$
< 0.12295. \qquad \text{(Calculated by computer)}
$$

Thus, the total variation distance between $\mathrm{Bin}(T, 1/6)$ and $\mathcal{N}(T/6, 5T/36)$ is at most $0.12295 + o(1)$, which implies the lemma. $\qquad \square$

Now we are ready to present the proof of Theorem D.2 (restated below).

**Theorem D.2.** *Under the setting with an unknown distribution of agent types, there exists a constant $\tau > 0$ such that no anonymous mechanism always outputs the alternative favored by more than half of the agents with probability more than $1 - \tau$ in any $\tau$-strong symmetric Bayes Nash equilibrium.*

*Proof.* We consider the following instance.

The prior distribution of the two worlds (world $L$ and world $H$) is given by $P_L = 0.98$ and $P_H = 0.02$. The probability distribution of the two signals under each of the two worlds is given by $P_{\ell L} = 1, P_{\ell H} = 0, P_{hL} = 5/6, P_{hH} = 1/6$. For each candidate-friendly agent $t$, we have $v_t(H, \mathbf{A}) = 3, v_t(H, \mathbf{R}) = 0, v_t(L, \mathbf{A}) = 2$ and $v_t(L, \mathbf{R}) = 1$. For each contingent agent $t$ we have $v_t(H, \mathbf{A}) = 3, v_t(H, \mathbf{R}) = 0, v_t(L, \mathbf{A}) = 1$ and $v_t(L, \mathbf{R}) = 2$. For each candidate-unfriendly agent $t$, we have $v_t(H, \mathbf{A}) = 1, v_t(H, \mathbf{R}) = 2, v_t(L, \mathbf{A}) = 0$ and $v_t(L, \mathbf{R}) = 3$. Lastly, $\mathcal{D}_{\Delta_3}$ is defined as follows: with probability $1/2$ we are in setting $X$ and the fractions of agents with types $F, C, U$ are $\alpha_F^{(1)} = 1/3, \alpha_C^{(1)} = 2/3, \alpha_U^{(1)} = 0$ respectively; with probability $1/2$ we are in setting $Y$, the fractions of agents with types $F, C, U$ are $\alpha_F^{(2)} = 0, \alpha_C^{(2)} = 1, \alpha_U^{(2)} = 0$ respectively. This finishes the description of the instance. Note there are 2 worlds and 2 settings yielding 4 possible environments which we label $LX, LY, HX$, and $HY$.

Let $\tau = 0.001$. Suppose there exists a mechanism $\mathcal{M}$ that outputs the majority wish with probability at least $1 - \tau$ in a symmetric strategy profile $\Sigma$. We will show that $\Sigma$ cannot be a $\tau$-strong Bayes Nash equilibrium.

Notice that the contingent agents are the majority in all the four settings. In both environments $HX$ and $HY$, each of which happens with probability $1\%$, the majority wish is always to accept. Thus, the mechanism must accept with probability at least $99\%$ in each environment for otherwise it will be far from being achieving $1 - \tau$ accuracy. Similarly, in both environments $LX$ and $LY$, each of which occurs with probability $49\%$, the mechanism must accept with probability at most $1\%$ for otherwise it will be far from being achieving $1 - \tau$ accuracy.

To show that $\Sigma$ cannot be a $\tau$-strong Bayes Nash equilibrium, consider that the set of deviating agents are all the candidate-friendly agents. Those candidate-friendly agents pretend they are contingent agents such that signal $\ell$ is received with probability $1/2$ and signal $h$ with probability $1/2$. These candidate-friendly agents will follow the strategy of the real contingent agents according to $\Sigma$. Let $\Sigma'$ be the resultant strategy profile. We aim to show that those deviating candidate-friendly agents can increase their utilities significantly in $\Sigma'$.

For an intuitive argument, in $LX$, the mechanism sees that all the agents are contingent, and the fraction of agents receiving signal $h$ follows distribution $\text{Bin}(T/3, 1/2)$; in $HY$, the mechanism also sees that all the agents are contingent, and the fraction of agents receiving signal $h$ follows distribution $\text{Bin}(T, 1/6)$. Lemma D.5 implies that the mechanism cannot distinguish between environments $LX$ and $HY$ with probability more than $87.7\%$. Before deviating, the mechanism will output $\mathbf{A}$ with probability at most $1\%$ in $LX$; after deviating, the mechanism will output $\mathbf{A}$ with probability at least $99\% \cdot 87.7\% > 1\%$ in $LX$ by confusing $LX$ with $HY$. The candidate-friendly agents will benefit from deviating.

To make the arguments in the previous paragraph more rigorous, let $\mathcal{M}(X) = 1$ if the mechanism outputs $\mathbf{A}$ when all the agents are contingent, agents play according to $\Sigma$, and the number of agents receiving signal $h$ is $X$. Let $\mathcal{M}(X) = 0$ if the output is $\mathbf{R}$ under the same circumstance. This definition is well-defined due to Proposition D.4.

In $HY$, the mechanism outputs $\mathbf{A}$ with probability $\mathbb{E}_{X \sim \text{Bin}(T, 1/6)}[\mathcal{M}(X)]$. In $LX$, when the candidate-friendly agents deviate to $\Sigma'$, the mechanism outputs $\mathbf{A}$ with probability $\mathbb{E}_{X \sim \text{Bin}(T/3, 1/2)}[\mathcal{M}(X)]$. Lemma D.5 implies

$$\left| \mathbb{E}_{X \sim \text{Bin}(T, 1/6)}[\mathcal{M}(X)] - \mathbb{E}_{X \sim \text{Bin}(T/3, 1/2)}[\mathcal{M}(X)] \right| < 0.123.$$

Since we have shown that the mechanism outputs $\mathbf{A}$ with probability at least $99\%$ in $HY$, the mechanism outputs $\mathbf{A}$ with probability at least $86.7\%$ in $LX$ when the candidate-friendly agents deviate. Since environment $LX$ happens with probability $0.49$, the expected utility for each candidate-friendly agent $t$ is at least $0.49 \times 86.7\% \times v_t(L, \mathbf{A}) = 0.84966$. However, without deviating, $\mathbf{A}$ will

be output with probability at most $0.98 \cdot 1\% + 0.02 = 0.0298$, and the expected utility for each candidate-friendly agent $t$ is upper-bounded by $0.0298 \times v_t(H, \mathbf{A}) = 0.0894$. We have seen that the candidate-friendly agents receive a utility gain of at least $0.76026 > \tau$. $\qquad\square$

As a remark, our impossibility result Theorem D.2 holds even for randomized mechanism. If mechanism can be randomized, Proposition D.4 becomes that $\{\alpha_F, \alpha_U, \alpha_C\} \cup \{\beta_{F,\ell}, \beta_{F,h}, \beta_{U,\ell}, \beta_{U,h}, \beta_{C,\ell}, \beta_{C,h}\}$ completely determines the *probability* that the mechanism output $\mathbf{A}$ (or $\mathbf{R}$). In the proof of Theorem D.2, $\mathcal{M}(X)$ becomes the *probability* that the mechanism output $\mathbf{A}$, rather than either 0 or 1. The remaining part of the proof is exactly the same.

# E  Extension to Non-binary Worlds and Signals

In Sect. E.1, we generalized the model to the setting with non-binary worlds and non-binary signals. In Sect. E.2, we present our mechanism for the setting with binary signals and non-binary worlds. In Sect. E.3, we show that the generalization to non-binary signals is straightforward.

## E.1  Model and Preliminaries

In our non-binary model, as in our binary model, $T$ agents are voting for two *alternatives*, $\mathbf{A}$ and $\mathbf{R}$ (corresponding to "accept" and "reject"). However, in our non-binary model there is a set of $N$ possible *worlds* (or *states*) $\mathcal{W} = \{1, \ldots, N\}$, where the higher value the more $\mathbf{A}$ is preferred to $\mathbf{R}$. Agents do not know which world is the actual world that they are in. They have a prior common belief on the likelihood of each world. Let $W$ be the actual world which is viewed as a random variable. Let $P_n = \Pr(W = n)$ be the prior over worlds. Each agent knows the values of $P_1, \ldots, P_n$ as prior beliefs. We further assume $P_n > 0$ for each $n$, for otherwise we can remove world $n$ from $\mathcal{W}$ without loss of generality.

Each agent will then receives a *signal* from the set $\mathcal{S} = \{1, \ldots, M\}$. Let $S_t$ be the random variable representing the signal that agent $t$ receives. Given $W = n$, for any $n$, the signals agents receive have the same distribution and are conditionally independent. Let $P_{mn} = \Pr(S_t = m \mid W = n)$ be the probability that signal $m$ will be received (by an arbitrary agent $t$) if the actual world is $n$. The set of values $\{P_{mn} : m = 1, \ldots, M; n = 1, \ldots, N\}$ is known by all the agents. Signals are positively correlated to the worlds:

$$\Pr(S_t \geq m \mid W = n_1) = \sum_{m'=m}^{M} P_{m'n_1} > \Pr(S_t \geq m \mid W = n_2) = \sum_{m'=m}^{M} P_{m'n_2} \quad (13)$$

for any worlds $n_1 > n_2$, any signal $m$, and any agent $t$.

The remaining definitions for the non-binary model in this section are rather analogous to the binary case, but we include them for completeness.

Each agent $t$ has a *utility function* $v_t : \mathcal{W} \times \{\mathbf{A}, \mathbf{R}\} \to \{0, 1, \ldots, B\}$. As mentioned earlier, a higher value of $W$ indicates $\mathbf{A}$ is more preferable: $v_t(n_1, \mathbf{A}) > v_t(n_2, \mathbf{A})$ and $v_t(n_1, \mathbf{R}) < v_t(n_2, \mathbf{R})$ for any $n_1, n_2 \in \mathcal{W}$ with $n_1 > n_2$. Since we can always rescale agents' utilities, for simplicity, we assume without loss of generality that agents' utilities are integers and bounded by $B \in \mathbb{Z}^+$. Agents, with their prior beliefs and receiving signals, will have posterior beliefs about the distribution of $W$ and react to the mechanism in a way maximizing their expected utilities.

We assume $v_t(n, \mathbf{A}) \neq v_t(n, \mathbf{R})$ for each agent $t$ and each $n \in \mathcal{W}$, so that agents always strictly prefer one alternative over the other. Given a world $n$, let $T(\mathbf{A}, n) = \{t \mid v_t(\mathbf{A}, n) > v_t(\mathbf{R}, n)\}$ be the set of agents that prefer $\mathbf{A}$ in world $n$ and let $\alpha_n^{\mathbf{A}} = \frac{|T(\mathbf{A},n)|}{|T|}$ be the fraction of agents that prefer $\mathbf{A}$ in world $n$. We can similarly define $T(\mathbf{R}, n)$ and $\alpha_n^{\mathbf{R}} = \frac{|T(\mathbf{R},n)|}{|T|} = 1 - \alpha_n^{\mathbf{A}}$. Naturally, $\alpha_n^{\mathbf{A}}$ is increasing in $n$ (when the underlying quality of the candidate increases, more agents prefer $\mathbf{A}$) and $\alpha_n^{\mathbf{R}}$ is decreasing in $n$. As before, we assume that the $\alpha_n^{\mathbf{A}}$ and $\alpha_n^{\mathbf{R}}$ are common knowledge, which is natural in many scenarios, including the faculty candidate hiring example. If this assumption does not hold, results in Sect. D show that we cannot achieve the truthful guarantee even under the setting $M = N = 2$.

For any world $n$, let

$$M(n) = \begin{cases} \mathbf{A} & \alpha_n^{\mathbf{A}} > \frac{1}{2} \\ \mathbf{R} & \text{otherwise} \end{cases}$$

be the majority preference if the actual world were $n$. We assume that $T$ is an odd number to avoid ties.

**Definition E.1.** Given a utility profile $\{v_1, \ldots, v_T\}$ and letting $n^*$ be the actual world, we say $M(n^*)$ is *the majority wish*.

The goal is to output the majority wish $M(n^*)$, the alternative that is preferred by at least half of the agents in the actual world.

Our results will sometimes require $T$, the number of agents, to be sufficiently large, and it may be helpful to think of $T \to \infty$. However, we will always assume that the parameters of the model: $B$, $\{P_n\}_{n \in \mathcal{W}}$, $\{P_{mn}\}_{m \in \mathcal{S}, n \in \mathcal{W}}$, and $\{\alpha_n^{\mathbf{A}}, \alpha_n^{\mathbf{R}}\}_{n \in \mathcal{W}}$, do not depend on $T$ in any way.

In the faculty candidate hiring example, the worlds $\mathcal{W} = \{1, \ldots, N\}$ describe the quality of the candidate, with 1 being the worst and $N$ being the best. The signals $S_t \in \{1, \ldots, M\}$ correspond to the impression of this candidate, with $S_t = 1$ being the worst impression and $S_t = M$ being the best impression. It is natural to assume that $S_t$'s are positively correlated to $W$, which agrees with our model.

### E.1.1 Candidate-Friendly, Contingent and Candidate-Unfriendly Agents

Let $\mathcal{L} = \{n \in \mathcal{W} \mid \alpha_n^{\mathbf{A}} < \frac{1}{2}\}$ and $\mathcal{H} = \{n \in \mathcal{W} \mid \alpha_n^{\mathbf{A}} > \frac{1}{2}\}$. Since we cannot have $\alpha_n^{\mathbf{A}} = \frac{1}{2}$ for an odd $T$, $\{\mathcal{L}, \mathcal{H}\}$ is a partition of $\mathcal{W}$. In addition, since $\alpha_n^{\mathbf{A}}$ is increasing in $n$, there exists a threshold such that all those $n$ below the threshold belong to $\mathcal{L}$ and all those $n$ above belong to $\mathcal{H}$. Indeed, $\mathcal{L}$ is the set of "low quality" worlds where $\mathbf{R}$ is preferred, and $\mathcal{H}$ is the set of "high quality" world is preferred.

For each agent $t$, define $\mathcal{L}_t = \{n \in \mathcal{W} \mid v_t(n, \mathbf{R}) > v_t(n, \mathbf{A})\}$ and $\mathcal{H}_t = \{n \in \mathcal{W} \mid v_t(n, \mathbf{A}) > v_t(n, \mathbf{R})\}$. Then $\mathcal{L}_t$ is the set of those low quality worlds based on agent $t$'s utility where $\mathbf{R}$ is preferred, and $\mathcal{H}_t$ is the set of those high quality worlds where $\mathbf{A}$ is preferred for $t$. Since $v_t(n, \mathbf{A}) - v_t(n, \mathbf{R})$ is increasing in $n$ (the first term is increasing and the second term is decreasing), each agent $t$ also has a personal threshold that separate $\mathcal{W}$ to $\mathcal{L}_t, \mathcal{H}_t$. We can define the candidate-friendly agents, contingent agents and candidate-unfriendly agents based on whether the personal threshold is below, equal to, or above the average threshold.

We say an agent $t$ is candidate-friendly if $\mathcal{H}_t \cap \mathcal{L} \neq \emptyset$. This says that there exists a world $n \in \mathcal{W}$ where the fraction of agents preferring $\mathbf{A}$ is below $1/2$ (i.e., $\alpha_n^{\mathbf{A}} < \frac{1}{2}$) but $t$ still prefers $\mathbf{A}$. Equivalently, an agent $t$ is candidate-friendly if $\mathcal{L}_t \subsetneq \mathcal{L}$, or $\mathcal{H} \subsetneq \mathcal{H}_t$. Correspondingly, an agent $t$ is candidate-unfriendly if $\mathcal{L}_t \cap \mathcal{H} \neq \emptyset$, or equivalently, $\mathcal{L} \subsetneq \mathcal{L}_t$, or $\mathcal{H}_t \subsetneq \mathcal{H}$. An agent $t$ is contingent if $\mathcal{L}_t = \mathcal{L}$, or equivalently, $\mathcal{H}_t = \mathcal{H}$. We still use $F, C, U$ to denote the three types of agents, and use $\alpha_F, \alpha_C, \alpha_U$ to denote their fractions as before. As a remark, a candidate-friendly agent (candidate-unfriendly agent resp.) does not always prefer $\mathbf{A}$ ($\mathbf{R}$ resp.) like before, (s)he merely has a threshold below (above resp.) the average.

Let $L = \max\{n \in \mathcal{L}\}$ be the maximum world where $\mathbf{R}$ is preferred by the majority, and $H = \min\{n \in \mathcal{H}\}$ be the minimum world where $\mathbf{A}$ is preferred by the majority. We clearly have $H = L + 1$. For each agent $t$, let $L_t = \max\{n \in \mathcal{L}_t\}$ be the maximum world where $\mathbf{R}$ is preferred, and let $H_t = \min\{n \in \mathcal{H}_t\}$ be the minimum world where $\mathbf{A}$ is preferred. Set $L_t = 0$ if $\mathcal{L}_t = \emptyset$ and $H_t = N + 1$ if $\mathcal{H}_t = \emptyset$. Clearly, $H_t = L_t + 1$.

### E.1.2 Additional Notations

Given a strategy profile $\Sigma = \{\sigma_1, \ldots, \sigma_T\}$ and a mechanism $\mathcal{M}$, let $\lambda_n^{\mathbf{A}, \mathcal{M}}(\Sigma)$ be the probability that alternative $\mathbf{A}$ is announced as the winner given the actual world is $n$, then $\lambda_n^{\mathbf{R}, \mathcal{M}}(\Sigma) = 1 - \lambda_n^{\mathbf{A}, \mathcal{M}}(\Sigma)$ is the probability that alternative $\mathbf{R}$ wins given the actual world is $n$. We will omit the superscript $\mathcal{M}$ when it is clear what mechanism we are discussing.

All the agents' *ex-ante* utilities depend exclusively on $\lambda_1^{\mathbf{A}}(\Sigma), \ldots, \lambda_N^{\mathbf{A}}(\Sigma)$ (or $\lambda_1^{\mathbf{R}}(\Sigma), \ldots, \lambda_N^{\mathbf{R}}(\Sigma)$), and each agent $t$'s utility is given by

$$u_t(\Sigma) = \sum_{n=1}^{N} P_n \left( \lambda_n^{\mathbf{A}}(\Sigma) v_t(n, \mathbf{A}) + \lambda_n^{\mathbf{R}}(\Sigma) v_t(n, \mathbf{R}) \right) \tag{14}$$

By substituting $\lambda_n^{\mathbf{R}}(\Sigma) = 1 - \lambda_n^{\mathbf{A}}(\Sigma)$,

$$u_t(\Sigma) = \sum_{n=1}^{N} P_n v_t(n, \mathbf{R}) + \sum_{n=1}^{N} P_n \lambda_n^{\mathbf{A}}(\Sigma)(v_t(n, \mathbf{A}) - v_t(n, \mathbf{R})). \tag{15}$$

We will always use $\Sigma^* = \{\sigma_1^*, \ldots, \sigma_T^*\}$ to denote the truthful strategy profile.

Table 1 lists all the frequently used notations.

| notation | meaning |
|---|---|
| $\mathcal{W} = \{1, \ldots, N\}$ | the set of all worlds |
| $\mathcal{S} = \{1, \ldots, M\}$ | the set of all signals |
| $P_n$ | the prior belief for the probability the actual world is $n$ |
| $P_{mn}$ | the probability of receiving signal $m$ under world $n$ |
| $v_t(n, \mathbf{A}), v_t(n, \mathbf{R})$ | the (*ex-post*) utility for agent $t$ for alternative $\mathbf{A}$, $\mathbf{R}$ if the actual world is $n$ |
| $u_t(\Sigma)$ | the (*ex-ante*) expected utility for agent $t$ given strategy profile $\Sigma$ |
| $\alpha_n^{\mathbf{A}}, \alpha_n^{\mathbf{R}}$ | the fraction of agents preferring $\mathbf{A}$, $\mathbf{R}$ under world $n$ |
| $M(n)$ | the majority favored alternative under world $n$ |
| $\mathcal{L}$ | the set of worlds where more than half of the agents prefer $\mathbf{R}$ |
| $\mathcal{H}$ | the set of worlds where more than half of the agents prefer $\mathbf{A}$ |
| $\mathcal{L}_t$ | the set of worlds where $\mathbf{R}$ is preferred for agent $t$ |
| $\mathcal{H}_t$ | the set of worlds where $\mathbf{A}$ is preferred for agent $t$ |
| $L$ | the maximum world where $\mathbf{R}$ is preferred by the majority |
| $H$ | the minimum world where $\mathbf{A}$ is preferred by the majority |
| $L_t$ | the maximum world where $\mathbf{R}$ is preferred for agent $t$ |
| $H_t$ | the minimum world where $\mathbf{A}$ is preferred for agent $t$ |
| $F, C, U$ | candidate-friendly agents, contingent agents, candidate-unfriendly agents |
| $\alpha_F, \alpha_C, \alpha_U$ | fractions of the three types of agents |
| $\lambda_n^{\mathbf{A}}(\Sigma), \lambda_n^{\mathbf{R}}(\Sigma)$ | the probability a given mechanism outputs $\mathbf{A}$, $\mathbf{R}$ for strategy profile $\Sigma$ |
| $\Sigma^*$ | the truthful strategy profile |

Table 1: Table of notations.

### E.2 Non-binary Worlds

In this section, we consider the generalization to the setting with more than two worlds $N > 2$, while keeping the binary signal assumption $M = 2$. We will see in the next section that the generalization to non-binary signals is simple. For this section, we will again use $\ell$ to denote signal 1 and $h$ to denote signal 2.

In the case $M = N = 2$, we have asked each agent his/her received signal, type, and posterior belief on the fraction of agents who will report signal 1. In the case $N > 2$ here, while it is still natural to ask an agent for his/her signal, asking for a posterior prediction and keeping the mechanism as before will not work here. In particular, this will make Theorem 3.2 fail. To reason this intuitively, it is easy to see that, if the mechanism is required to output the alternative favored by the majority, the mechanism will output $\mathbf{A}$ if the actual world is in $\mathcal{H}$ and output $\mathbf{R}$ if the actual world is in $\mathcal{L}$. To ensure this, we need to make sure the median of the posterior prediction is between $P_{hL}$ and $P_{hH}$. However, while this is true for $N = 2$ as Theorem 3.1 suggests (in fact, all the possible posterior predictions, $T_{h\ell}$ and $T_{hh}$, are between $P_{hL}$ and $P_{hH}$), this is not necessarily true for $N > 2$.

As a solution to this issue, for each contingent agent $t$, we ask him/her for a value between $P_{hL_t}$ and $P_{hH_t}$ (agents with $\mathcal{L}_t = \emptyset$ report a value between 0 and $P_{h1}$ and agents with $\mathcal{H}_t = \emptyset$ report a value between $P_{hN}$ and 1), and the median of these values will be between $P_{hL}$ and $P_{hH}$. A natural way to ask an agent for this value can be, *please give a percentage value q such that you would like*

*alternative **A** if the fraction of agents reporting signal h is more than q percent, and you would like alternative **R** otherwise.*

Our mechanism is presented in Mechanism 3.[7]

---

**Mechanism 3** The Wisdom-of-the-Crowd-Voting Mechanism for $N > 2$ and $M = 2$

---

1: Each agent $t$ reports to the mechanism the signal (s)he receives (either $\ell$ or $h$), the type (either $F$, $C$, or $U$).
2: If agent $t$ reports type $F$, his/her reported signal will be automatically treated as $h$; if agent $t$ reports type $U$, his reported signal will be automatically treated as $\ell$.
3: If an agent reports type $C$, ask him/her to report a value $q_t \in [0, 1]$ such that (s)he would like **A** *if and only if* the fraction of agents who report $h$ is more than $q_t$. *The value $q_t$ should be given with the treatment in the previous step being considered, and the mechanism makes this clear to the agents.* For an agent with type $F$, set $q_t = 0$. For an agent with type $U$, set $q_t = 1$.
4: Compute the *median* of those $q_t$, denoted by $\bar{q}$.
5: If the fraction of the agents reporting $h$ is more than the median $\bar{q}$, announce **A** being the winning alternative; otherwise, announce **R** being the winning alternative.

---

In our faculty candidate hiring example, the questionnaire corresponding to the mechanism looks like the following:

1. Choose one of the following: as compared with the average faculty member, independent of the candidate's qualification:

    I  I am more predisposed toward rejection;

    II  I am more predisposed toward acceptance;

    III  Neither I nor II, i.e., I am with the average faculty member.

2. What is your impression of this candidate during the individual interview between you and this candidate?

    (a)  I had a good impression.

    (b)  I did not have a good impression.

3. Your "provisional ballot" will be cast as follows:

    - Accept: if you choose II in Question 1, or, if you choose III in Question 1 and (a) in Question 2;
    - Reject: if you choose I in Question 1, or, if you choose III in Question 1 and (b) in Question 2.

4. If you answered III for Question 1, what fraction of provisional ballots do you predict will be cast for accept?

Next, we will prove that the mechanism outputs the alternative favored by the majority with high probability, and the truthful strategy profile is an $\varepsilon$-strong Bayes Nash equilibrium for $\varepsilon = o(1)$. For simplicity and clarity in describing the ideas behind the proofs, we will not perform the Chernoff bound analyses as in Section B.3, and we will assume $T \to \infty$ as it is in Sect. 3. As a result, for any world $n$, the fractions of agents receiving signal $\ell$ and $h$ are, almost surely, $P_{\ell n}$ and $P_{hn}$ respectively.

**Theorem E.2.** *Suppose $T \to \infty$. If all the agents play the truthful strategy, then our mechanism outputs an alternative favored by more than half of the agents.*

*Proof.* Suppose all the agents report truthfully. Let $n$ be the actual world. We need to show that **A** is announced if and only if $n \in \mathcal{H}$. We assume $n \in \mathcal{H}$ without loss of generality, as the analysis for $n \in \mathcal{L}$ is similar. When all the agents play the truthful strategy profile $\Sigma^*$, the fraction of agents reporting signal $h$ is $\alpha_F + \alpha_C \cdot P_{hn}$.

On the other hand, for any agent $t$, (s)he prefers **A** if $n \in \mathcal{H}_t$, and (s)he prefers **R** if $n \in \mathcal{L}_t$. If (s)he were asked to give a value such that (s)he would like **A** if and only if the fraction of agents who *receive* signal $h$ is less than this value, (s)he would have report a value between $P_{hL_t}$

---

[7]We can consider the same modification as in the binary case (Section C).

and $P_{hH_t}$. Now, considering that, as instructed by the mechanism, those $\alpha_F \cdot T$ (resp. $\alpha_U \cdot T$) agents will always *report* signal $h$ (resp. signal $\ell$) regardless of what they receive, (s)he will report $q_t \in (\alpha_F + \alpha_C \cdot P_{hL_t}, \alpha_F + \alpha_C \cdot P_{hH_t})$.

It is then easy to see that the median $\bar{q}$ is between $\alpha_F + \alpha_C \cdot P_{hL}$ and $\alpha_F + \alpha_C \cdot P_{hH}$, which is less than the fraction of agents reporting signal $h$ (which is $\alpha_F + \alpha_C \cdot P_{hn}$ as computed earlier). The last step of our mechanism will make sure **A** is output. □

As a remark, if we do not assume $T \to \infty$, to show that the statement in Theorem E.2 fails with an exponentially low probability, we need to make an extra assumption that the median $\bar{q}$ is not exponentially close to the two endpoints $\alpha_F + \alpha_C \cdot P_{hL}$ and $\alpha_F + \alpha_C \cdot P_{hH}$. This is a natural assumption, as an agent's reported value should not depend on $T$. In addition, in practice, it is natural to expect that most agents will report values that are around the midpoint of the interval $(\alpha_F + \alpha_C \cdot P_{hL_t}, \alpha_F + \alpha_C \cdot P_{hH_t})$.

Next, we show that our mechanism satisfies the truthful property. Again, we consider $T \to \infty$.

**Theorem E.3.** *Suppose $T \to \infty$. The truthful strategy profile form a strong Bayes Nash Equilibrium.*

The ideas behind the proof of Theorem E.3 is similar as before. Firstly, those contingent agents will not deviate from the truthful strategy, as their utilities have already been maximized. Secondly, we can prove a lemma similar to Lemma 3.5 showing that there is a conflict of interest between an arbitrary candidate-friendly agent and an arbitrary candidate-unfriendly agent. This shows that the set of deviating agents $D$ can only contain either candidate-friendly agents or candidate-unfriendly agents. This further implies that more than half of the agents are truth-telling. Finally, the use of median in our mechanism ensures that less than half of the agents' deviating cannot change the output alternative.

**Lemma E.4.** *Let $\Sigma^*$ be the truthful strategy profile and $\Sigma'$ be an arbitrary strategy profile. Let $t_1$ be an arbitrary candidate-friendly agent and $t_2$ be an arbitrary candidate-unfriendly agent. Suppose $T \to \infty$. We have*

1. *If $u_{t_1}(\Sigma') > u_{t_1}(\Sigma^*)$, then $u_{t_2}(\Sigma') < u_{t_2}(\Sigma^*)$.*

2. *If $u_{t_1}(\Sigma') < u_{t_1}(\Sigma^*)$, then $u_{t_2}(\Sigma') > u_{t_2}(\Sigma^*)$*

The proof of this lemma is more involved than that of Lemma 3.5, but the ideas behind are similar.

*Proof.* We will only prove (1), as the proof for (2) is similar. For ease of notation, in this proof, we assume $t_1 = 1$ and $t_2 = 2$ without loss of generality. Suppose $u_1(\Sigma') > u_1(\Sigma^*)$, and we aim to show $u_2(\Sigma') < u_2(\Sigma^*)$. Theorem E.2 implies that $\lambda_n(\Sigma^*) = 0$ for all $n \in \mathcal{L}$ and $\lambda_n(\Sigma^*) = 1$ for all $n \in \mathcal{H}$. Firstly, we show that

$$\sum_{n \in \mathcal{L} \cap \mathcal{H}_1} P_n \left( \lambda_n^{\mathbf{A}}(\Sigma') - \lambda_n^{\mathbf{A}}(\Sigma^*) \right) > \sum_{n \in \mathcal{H} \cap \mathcal{L}_2} P_n \left( \lambda_n^{\mathbf{A}}(\Sigma^*) - \lambda_n^{\mathbf{A}}(\Sigma') \right). \tag{16}$$

This is because

$$0 < u_1(\Sigma') - u_1(\Sigma^*) \qquad \text{(by our assumption)}$$

$$= \sum_{n=1}^{N} P_n \left( \lambda_n^{\mathbf{A}}(\Sigma') - \lambda_n^{\mathbf{A}}(\Sigma^*) \right) (v_1(n, \mathbf{A}) - v_1(n, \mathbf{R})) \qquad \text{(by (15))}$$

$$\leq \sum_{n \in \mathcal{W} \backslash (\mathcal{L}_1 \cup \mathcal{H}_2)} P_n \left( \lambda_n^{\mathbf{A}}(\Sigma') - \lambda_n^{\mathbf{A}}(\Sigma^*) \right) (v_1(n, \mathbf{A}) - v_1(n, \mathbf{R})) \qquad (\dagger)$$

$$= \sum_{n \in \mathcal{L} \cap \mathcal{H}_1} P_n \left( \lambda_n^{\mathbf{A}}(\Sigma') - \lambda_n^{\mathbf{A}}(\Sigma^*) \right) (v_1(n, \mathbf{A}) - v_1(n, \mathbf{R}))$$

$$- \sum_{n \in \mathcal{H} \cap \mathcal{L}_2} P_n \left( \lambda_n^{\mathbf{A}}(\Sigma^*) - \lambda_n^{\mathbf{A}}(\Sigma') \right) (v_1(n, \mathbf{A}) - v_1(n, \mathbf{R}))$$

$$\leq \sum_{n \in \mathcal{L} \cap \mathcal{H}_1} P_n \left( \lambda_n^{\mathbf{A}}(\Sigma') - \lambda_n^{\mathbf{A}}(\Sigma^*) \right) (v_1(L, \mathbf{A}) - v_1(L, \mathbf{R}))$$

$$- \sum_{n \in \mathcal{H} \cap \mathcal{L}_2} P_n \left( \lambda_n^{\mathbf{A}}(\Sigma^*) - \lambda_n^{\mathbf{A}}(\Sigma') \right) (v_1(L, \mathbf{A}) - v_1(L, \mathbf{R})), \qquad (\ddagger)$$

which implies (16), where both Step ($\dagger$) and ($\ddagger$) are based on the following facts. In particular, ($\dagger$) is based on the first two facts, and ($\ddagger$) is based on the first and the third facts.

- for $n \in \mathcal{L}$, $\lambda_n^{\mathbf{A}}(\Sigma') - \lambda_n^{\mathbf{A}}(\Sigma^*) = \lambda_n^{\mathbf{A}}(\Sigma') - 0 \geq 0$; for $n \in \mathcal{H}$, $\lambda_n^{\mathbf{A}}(\Sigma^*) - \lambda_n^{\mathbf{A}}(\Sigma') = 1 - \lambda_n^{\mathbf{A}}(\Sigma') \geq 0$.

- $v_1(n, \mathbf{A}) - v_1(n, \mathbf{R})$ is negative for $n \in \mathcal{L}_1$ and is positive for $n \in \mathcal{H}_2$. Notice that this is also true for $v_2$.

- the expression $v_1(n, \mathbf{A}) - v_1(n, \mathbf{R})$ is increasing in $n$. This is true for any agent $t$. In particular, for each agent $t$, $v_t(n, \mathbf{A})$ is increasing in $n$ and $v_t(n, \mathbf{R})$ is decreasing in $n$.

Next, we show that (16) implies $u_2(\Sigma') < u_2(\Sigma^*)$. By the same calculations and analyses above, we have

$$u_2(\Sigma') - u_2(\Sigma^*) \leq \sum_{n \in \mathcal{L} \cap \mathcal{H}_1} P_n \left( \lambda_n^{\mathbf{A}}(\Sigma') - \lambda_n^{\mathbf{A}}(\Sigma^*) \right) (v_2(L, \mathbf{A}) - v_2(L, \mathbf{R}))$$

$$- \sum_{n \in \mathcal{H} \cap \mathcal{L}_2} P_n \left( \lambda_n^{\mathbf{A}}(\Sigma^*) - \lambda_n^{\mathbf{A}}(\Sigma') \right) (v_2(L, \mathbf{A}) - v_2(L, \mathbf{R}))$$

$$\text{(same calculations above)}$$

$$< 0, \qquad \text{(by } v_2(L, \mathbf{A}) - v_2(L, \mathbf{R}) < 0 \text{ and (16))}$$

which implies the lemma. $\qquad \square$

Now we are ready to prove Theorem E.3.

*Proof of Theorem E.3.* Suppose otherwise and there is a set of deviating agents $D$. Let $\Sigma'$ be the profile after the deviation of agents in $D$. Firstly, we show that $D$ cannot contain a contingent agent. Notice that such an agent's utility has already been maximized by the truthful profile $\Sigma^*$. Suppose $\lambda_n^{\mathbf{A}}(\Sigma') \neq \lambda_n^{\mathbf{A}}(\Sigma^*)$ for certain $n$. It must be that $\lambda_n^{\mathbf{A}}(\Sigma') > \lambda_n^{\mathbf{A}}(\Sigma^*) = 0$ if $n \in \mathcal{L}$, and $\lambda_n^{\mathbf{A}}(\Sigma') < \lambda_n^{\mathbf{A}}(\Sigma^*) = 1$ if $n \in \mathcal{H}$. It is then easy to see that this agent's utility will decrease, which contradicts to 2 of Definition 2.1. Suppose $\lambda_n^{\mathbf{A}}(\Sigma') = \lambda_n^{\mathbf{A}}(\Sigma^*)$ for all $n$. We have $u_t(\Sigma') = u_t(\Sigma^*)$ for every agent $t$. This already contradicts to 3 of Definition 2.1.

Next, Lemma E.4 ensures that $D$ cannot contain both a candidate-friendly agent and a candidate-unfriendly agent. Assume without loss of generality that $D$ only contains candidate-friendly agents. In order to maximize the chance that $\mathbf{A}$ is output, those candidate-friendly agents need to maximize the fraction of agents reporting signal $h$ and minimize the median $\bar{q}$. However, the mechanism always does this for them in the truth-telling profile $\Sigma^*$: Step 2 makes sure they report signal $h$, and Step 3 makes sure they report $q_t = 0$. Therefore, those candidate-friendly agents' utilities are maximized by truth-telling, which contradicts to our assumption for $D$. $\qquad \square$

### E.3 Non-binary Signals

There is a simple reduction from the non-binary signal setting to the binary-signal setting. Suppose the signal space is $\{1, \ldots, M\}$. To reduce it to a binary signal space $\{\ell, h\}$, we set an arbitrary non-integer number $s_\top$ between 1 and $M$. All the signals less than $s_\top$ are reduced to the "bad" signal $\ell$, and all the signals greater than $s_\top$ are reduced to the "good" signal $h$. The mechanisms in the previous sections can be adapted to the setting here. The mechanisms are the same as before, except for the following change: whenever the mechanism asks an agent for a binary signal in the previous setting, the mechanism asks the agent whether the signal (s)he received is less than or more than $s_\top$, which corresponds to signal $\ell$ and $h$ respectively.

For the mechanism in Section E.2, all the properties, including that the mechanism outputs the alternative favored by more than half of the agents and that the truth-telling strategy profile form a strong Bayes Nash Equilibrium, continue to hold in the non-binary signal setting with exactly the same proofs.

If we are dealing with binary world $\mathcal{W} = \{L, H\}$, for the mechanism in Section 3, it is easy to see that these properties also continue to hold here if we prove the following inequality that is similar to the one in Theorem 3.1:

$$\forall m \in \{1, \ldots, M\} : P_{hH} > T_{hm} > P_{hL}, \tag{17}$$

where $P_{hL}$ and $P_{hH}$ are the probabilities that a signal above $s_\top$ is received if the actual world is $L$ and $H$ respectively, and $T_{hm}$ is the probability that an agent who receives signal $m$ believes that another agent will receive a signal that is more than $s_\top$. Intuitively, if (17) holds, all the agents' posterior predictions are still between $P_{hL}$ and $P_{hH}$, and the majority wish will still be "surprisingly popular". The proof of (17) is by straightforward Bayesian analysis, and is left to the readers.