# OpenReview forum: "Wisdom of the Crowd Voting: Truthful Aggregation of Voter Information and Preferences"
_NeurIPS.cc/2021/Conference — NeurIPS 2021 Poster_

### Official Review · Reviewer_M3RC · 2021-07-13

**Rating:** 6
**Confidence:** 4

**Summary:**

The paper considers two alternative elections in which the voters are imperfectly informed. There is an unobservable state that may affect some voters’ preferences. Each voter receives a signal correlated with the state. The voter’s signals are conditionally independent and follow the same publicly known distribution. The designer’s goal is to elicit truthful information and aggregate the information to find the alternative that is preferred by the majority. The paper proposes a mechanism that guarantees truthful reporting is an approximate strong BNE and outputs the right alternative with high probability.


**Limitations And Societal Impact:**

Yes.

**Main Review:**

The problem considers an interesting problem. The idea of combining information elicitation and voting when the voters are imperfectly informed is good. The problem setup as well as the main technique builds heavily on the previous work [Prelec et al. 2017]. But the trick of using the median to guarantee strong truthfulness is cute.

The overall writing is good, but the definition of the state in the model is a little bit confusing (see the comments below). I did not find any glaring mistakes in the proofs.

Comments:

The way that the state W is introduced in the model is confusing. “Here, L stands for low quality where more agents prefer R, and H stands for high quality where more agents prefer A.” The state W seems to be defined in a way that depends on the voting outcome, but the voting outcome depends on the voters’ preference which depends on the state. Suppose there are 30% of rejecting voters, 30% of accepting voters, and 40% of swing voters. Then should the state be L or H in this case? Assuming it’s L, then the 40% swing voters would prefer rejecting and it is indeed L. Assuming it’s H, then the swing voters would prefer accepting and it is indeed H. It seems that the state should be a predetermined random variable that does not depend on the outcome.

It would be good to discuss a little bit how to extend the result to more general settings in the main text.

Questions:

Please comment on the definition of the state. Does it have to represent which alternative is more preferred? Is it actually considered as a predetermined random variable that is independent of the outcome?

At Line 369: “We have seen that the deviating coalition D cannot contain any contingent agents (since their utilities have already been maximized)” Is this utility the ex-ante utility or the ex-post utility?


**Time Spent Reviewing:**

5

---

> ### Author Response · Authors · 2021-08-07
> **Response to Reviewer M3RC**
>
> Thanks very much for your careful review!
>
> $W$ is merely a random variable for the actual underlying world, which does not depend on the voting outcome. “$W = L$” describes the event that the actual world is $L$, and “$W = H$” describes the event that the actual world is $H$. So the intuition is that, if the candidate is of low quality ($W = L$), then fewer agents will want (not based on votes but based on underlying utilities) to hire the candidate than if the candidate is of high quality ($W = H$).
>
> In fact, we have only used $W$ in the context where we want to describe these two events (the only part where we use $W$ in the main text is where we are defining $P_L$ and $P_H$, where we have written $P_L = \Pr(W = L)$ and $P_H = \Pr(W = H)$). We will find a better way to make the notation less confusing. Maybe just removing the notation of $W$.
>
> Line 369: we are talking about the ex-ante utilities.
>
> The mechanism for the general non-binary setting also uses the median trick, but it does not rely on the surprisingly popular method at all. We will try to put a brief discussion on the technical ideas in the main text in the final version of this paper.

---

> > ### Comment · Reviewer_M3RC · 2021-08-29
> > **Thanks for the reply**
> >
> > Thanks for the reply and the clarifications. I have no further questions about the paper.

---

### Official Review · Reviewer_k218 · 2021-07-15

**Rating:** 7
**Confidence:** 3

**Summary:**

This paper considers the two-alternative voting problem with imperfectly informed voters. In this setting, each voter receives a signal which is positively correlated to voter's true preferences (over two alternatives). The goal is to aggregate these signals to find out the majority voting winner w.h.p. Specifically, the paper gives a mechanism such that  1) the mechanism will output the majority voting winner under the true preferences w.h.p.  2) the mechanism incentives every voter to truthfully report the signal they received.

The main technical contribution of this work is that their mechanism uses the idea of "surprisingly popular" approach (Prelec et al., 2017)  from the information aggregation literature. They show that "surprisingly popular" approach can be well adapted to two-alternative social choice setting and can incentivize voters to report truthfully (for two-alternative). In addition, they show their mechanism is user-friendly, and does not require voters to understand deep mathematics.

**Limitations And Societal Impact:**

Authors have adequately addressed the limitations, and I don't see any potential negative societal impact of this work.

**Main Review:**

The problem this paper tries to solve is well-motivated, and I believe it is a very important problem. Social choice problems with imperfectly informed voters are practical and significant in real world. It's actually a game-theoretic version of information aggregation, they not only need to aggregate partial information to get the correct answer w.h.p, but they also have to guarantee that nobody is willing to strategically misreport information they know. Previous work in the information aggregation literature does not consider potential strategic behaviors of participants. This paper is novel as it is the first one considers both correctness and truthfulness in this setting.

The major contribution of this work is extending the "surprisingly popular" approach (Prelec et al., 2017) to the social choice setting. In their mechanism, each voter are asked to provide the signal they received as well as their prediction of other voters' preferences. They prove this mechanism is both correct and truthful. I really appreciate that authors use a paragraph to describe how to design questionnaire for this mechanism, which clearly demonstrates that this mechanism is user-friendly and easy to implement.

The weakness of this paper is that the two-alternative social choice setting is too simple. Moreover, by reading the proof,  I might be missing something but I have a sense that in this setting, a mechanism is truthful as long as it is correct. So I guess maybe plugging any correct information aggregation mechanism other than the "surprisingly popular" mechanism will result in a truthful mechanism as well. In that case, two-alternative voting with imperfectly informed voters is essentially the same as conventional information aggregation setting.

Overall, I think it's a good paper which is well-motivated and clearly written. However, the setting seems to be too weak and I feel it's just a somewhat straightforward extension of the existing "surprisingly popular" mechanism (while it is still valuable to point out the connection between two settings).

### Questions:
1) What if I replace median in the mechanism with mean? Is there any obvious objection to that?
2) Is it true that truthfulness ($\epsilon$-strong BNE) is directly followed by correctness (output the correct answer w.h.p)? Is it true that plugging any (non game-theoretic) mechanism from the information aggregation literature will immediately result in a truthful voting scheme in your setting?

### Typos:
1) line 180 "$v_t(H, A) > v_t(L, A)$" -> "$v_t(H, A) > v_t(H, R)$"

**Time Spent Reviewing:**

4

---

> ### Author Response · Authors · 2021-08-07
> **Response to Reviewer k218**
>
> Thanks very much for your careful review!
>
> Question 1:
> Both the mean and the median can guarantee the correctness. However, to guarantee truthfulness, it is essential that the median is used instead of the mean. Intuitively, in the case where there are less than 50% of the candidate-friendly agents and there are less than 50% of the candidate-unfriendly agents, we need to make sure the aggregated prediction is located among those contingent agents’ predictions (so that the surprisingly popular method is still functional), and it is the median, not the mean, that can guarantee this. To see a concrete example, suppose there are 49% of the candidate-friendly agents and 51% of the contingent agents (there are no candidate-unfriendly agents in particular). Suppose $h$ is received in world $L$ with probability 0.95 (i.e., $P_{hL} = 0.95$) and $h$ is received in world $H$ with probability 0.97 (i.e., $P_{hH} = 0.97$). Those contingent agents’ predictions will be between 0.95 and 0.97. If $L$ is the actual world, 95% of the agents receive signal $h$, and if $H$ is the actual word, 97% of the agents receive signal $h$. If we take the mean instead of the median, those candidate-friendly agents will pretend they are contingent agents (by misreporting their type to C), and report prediction 0% and signal $h$. By doing this, the mean of the prediction will be dragged to a value around 50% (since those 49% of the agents report 0%, and those 51% of the agents report values between 95% and 97%), in which case signal $h$ is always surprisingly popular (as both 95% and 97% are much larger than a number around 50%). Thus, those candidate-friendly agents benefit from this misreporting, especially when the prior probability of world $L$, i.e., $P_L$, is high. In this example, if we use the median, the median of the predictions is still between 95% and 97%, and the surprisingly popular method still works.
>
> Question 2:
> As we have addressed in the response in Question 1, the mechanism needs to be designed in a very careful way to achieve truthfulness. Using the previous example, if we were to use the mean to aggregate the predictions, the truthfulness cannot be guaranteed. However, both the mean and the median can guarantee the correctness. In particular, we have many previous failed attempts to make the mechanism truthful before we came up with Mechanism 1.
>
> “two-alternative social choice setting is too simple.”
> We see the simplicity of our work (and any work really) as a HUGE PLUS.   The simplicity means it should be implementable. This could be huge.  We cannot recall the last time someone suggested a voting scheme for two alternatives that is fundamentally different from the majority vote.  Most of the well-known mechanisms that have a big impact are those that are simple enough to appear in undergraduate textbooks.  We believe this is the type of big impact paper that NeurIPS is looking for.
>
> Moreover, the design of our mechanism may be simple, but it is not straightforward. Indeed, to the best of our knowledge, it is the first application of “surprisingly popular” to social choice in the 16 years since Bayesian Truth Serum (BTS) came out (Prelec, Seung, and McCoy’s surprisingly popular method can be seen as a simple application of BTS, which originally defined “surprisingly popular”).
>
> This also might be a critique like the first reviewer that the two-alternative setting is “somewhat narrow” in its application.  Please see the response to Reviewer 1.
>
> Thanks a lot for catching up the typo at Line 180.

---

> > ### Comment · Reviewer_k218 · 2021-08-30
> > **Update**
> >
> > Thanks for your response and clarification. The response makes it clear to me that the extension of the "surprisingly popular" scheme in this setting is non-trivial. I will keep my score.

---

### Official Review · Reviewer_3nmy · 2021-07-16

**Rating:** 6
**Confidence:** 3

**Summary:**

The authors consider elections that have two alternatives and three distinct classes of voters: candidate-friendly, candidate-unfriendly, and contingent agents. Candidate-friendly and candidate-unfriendly agents have made up their minds before the election, but each contingent voter's opinion depends on an unobservable state variable; they each receive a private signal about this state variable. The authors present a mechanism that elicits private signals from voters and which outputs the correct answer based on the state of the world. This mechanism is also strategyproof against even coalitions in the sense that truthful reporting is a strong Bayes Nash equilibrium. The mechanism draws on the "surprisingly popular" method of Prelec et al.


**Ethical Concerns:**

None.

**Limitations And Societal Impact:**

Yes.

**Main Review:**

I found the setting to be a fairly natural one -- at least in US politics, it seems to fit well. Furthermore, I enjoyed the incorporation of the "surprisingly popular" mechanism, which is a very elegant and nice result.

The technical results are not too surprising, and they use mostly standard techniques (at least in the body of the paper).The proofs are relatively cleanly laid out and it is easy to follow the big ideas in the paper's technical content.

If I'm not mistaken, the crux of the correctness proof relies on the chain of inequalities in Theorem 3.1. In particular, this means that the population breakdown between candidate-friendly, candidate-unfriendly, and contingent agents must be exactly known public information, which seems like a very strong assumption. I assume that imperfect knowledge of this breakdown (however slight the error) could, in theory, lead to the population making the wrong decision.

Also, the Bayesian update step is quite complicated, and I don't think people will perfectly update every time.

I think this paper would significantly benefit from some experiments in which people are assigned affiliations (candidate-friendly, candidate-unfriendly, and contingent), and contingent agents must report the fraction of contingent agents who will report signal h. If this works well in practice even when (a) agents don't perfectly perform Bayesian updates and/or (b) the exact population breakdown isn't known, then that would be a very nice proof of concept.

Minor comments:

All instances of "surprising popular" should be surprisingly popular" (e.g., 90, 125, 253, ...)
124: suggest removing "rely on the same notions"
Thm 3.1: period at end
279: You should note that c as defined after line 279 is for Chernoff bounds later (e.g., in Thm 3.2).

**Time Spent Reviewing:**

2

---

> ### Author Response · Authors · 2021-08-07
> **Response to Reviewer 3nmy**
>
> Thanks very much for your thoughtful review.
>
> Public Knowledge of Prior:
>
> The assumption that “the population breakdown between candidate-friendly, candidate-unfriendly, and contingent agents must be exactly known public information” is natural in many scenarios (e.g., in a CS department, the fraction of theory, AI, software, hardware faculty members are common knowledge), and it is also essential. Without this, we have a strong impossibility result shown in Theorem E.2 and Corollary E.3 in Appendix E.
>
> Also, we note that the mechanism itself does not require any such knowledge.
>
> Bayesian Updating of Agents:
>
> We have remarked about the Bayesian update in the first paragraph of Sect. 4. Although the Bayesian assumption may not be completely realistic in practice, we believe the theoretical properties of our mechanism still hold to a certain extent. For example, agents may not exactly predict $\alpha_CT_{hh}+\alpha_F$ or $\alpha_CT_{h\ell}+\alpha_F$ in practice, but it is reasonable to assume that their predictions are roughly around these two numbers, or in between. If so, all the theoretical properties will still hold. Prelec et al.’s work also assumes Bayesian agents in the theoretical analysis of the surprisingly popular method, but their empirical experiments with human subjects suggest the method still works in practice.
>
> Experiments:
>
> We agree that performing experiments is a great future direction, and we mentioned this in Sect. 4 as well.  However, running experiments may not be quite as easy as the review seems to imply.  The reason is that just assigning people roles in an artificially constructed setting is unlikely to lead to the intuitive reasoning that our mechanism relies on.  In real situations, it seems (from prior work) that people are able to naturally perform these updates. It seems much less likely that MTurkers in contrived situations where they are simply given the distributions described mathematically will update correctly.  Thus, such experiments need to be carefully reasoned about (we are attempting this reasoning now) to set up a reliable test for this mechanism and is beyond the scope of this work.
>
> We hope that our response will enable you to raise your score.

---

### Official Review · Reviewer_kct9 · 2021-07-17

**Rating:** 5
**Confidence:** 3

**Summary:**

This paper studies a setting that combines social choice and information aggregation.  As in a classic social choice problem, individuals with preferences want to choose between two options, but as in an information aggregation setting they do not have the knowledge to know exactly what their preferences are, instead having only a signal correlated with the true state of the world.  The paper introduces a “Wisdom-of-the-Crowd-Voting” Mechanism, which implements the majority wish in an (approximate) strong Bayes Nash equilibrium.

**Limitations And Societal Impact:**

On the negative side, the results are somewhat narrow as the approach would start to run into all the standard challenges in social choice once there are more than two alternatives.  There is also a bit of a tension where the solution concept aims for ex-ante properties (e.g. Bayes-Nash), but somewhat oddly the goal of the algorithm is ex-post (choosing the preferred alternative conditioned on the true state).  This is perhaps defensible, but certainly seems different from the more common goal of maximizing expected utility.  In particular, this mechanism doesn’t seem appealing in the finite T case to agents who have, e.g., a weak preference to avoid false negative errors but a strong preference to avoid false positives.

I also have a number of comments on the presentation, but they largely seem easy to resolve, although the page limit seems to be a bit of a constraint and some adjustments may be needed elsewhere:
- As someone not familiar with many of the cited papers I found 1.2 (and to a lesser extent 1.3) hard to follow.  Eventually I just gave up and read the introduction of Austen-Smith and Banks, after which it made more sense. So more is needed here to make this material self-contained.
- I’m much more familiar with the work discussed in Appendix B and was wondering by the end of the paper why none of it was discussed.  I think this really needs to be part of the main paper to position it properly and putting it in the appendix is at least somewhat abusive of the page limit.
- I disagree with some of the discussion around the revelation principle.  While I agree that this mechanism is not quite the revelation principle applied to plurality voting, it still seems to me to be an application of it.  In particular, the revelation principle does not actually require the entire agent’s type to be revealed, but only a sufficient statistic of it to act on the agent’s behalf which is what this mechanism appears to do.  Maybe there is a deeper reason this interpretation is wrong, but the superficial one presented isn't convincing.
- To what extent is Theorem 3.1 novel?  It seems to be analyzing Prelec et al. but translated to the model of this paper so I’m not clear whether it is just a translation of their analysis or a new analysis of their algorithm that extends to this one?
- The transition into 3.2 seems rushed and jumps into the analysis in mid thought: “At Step 3 of the mechanism...”

**Main Review:**

The mechanism appears novel, natural, and reasonably practical, as it asks agents only to report their ordinal utilities and beliefs about others conditional on their signal.  (In the spirit of ideas from peer prediction, I think we could even dispense with the reported belief if favor of the signal since one can be inferred directly from the other when the signal structure is known to the mechanism designer.)  The properties established for it are satisfying.

Post Author Response:
Thanks for your response.  On a few of the points raised:

Two alternatives - there is certainly plenty of work that looks at settings with two outcomes, but I still do think it is fair to call this restriction a limitation of the results in the paper.

Ex-ante/Ex-post and Finite T - It seems like there are some worthwhile things to say here, and it would be good for some of them to make it into the paper.  I will point out though, that these issues aren't obvious to me.  For example, this paper already makes "strong additional assumptions" by imposing the signal structure and this also makes me question whether majority vote and variants really are the only real alternative.  It would strengthen the paper if theoretical results along these lines could be established.

Theorem 3.1 - Even in the context of the section as review, it still doesn't read as obvious to me that this analysis is direct from them - if it is, why is there a complete proof and so much discussion of how to prove it?  So I think at least a bit more guidance for the reader about where the ideas are coming from and why they are being presented in the detail they are is needed.

**Time Spent Reviewing:**

2

---

> ### Author Response · Authors · 2021-08-07
> **Response to Reviewer kct9**
>
> Thanks very much for your thoughtful review!
>
> Let us respond to a few points that you made where we disagree or would like to clarify:
>
> “Results are somewhat narrow”:  There are endless papers in top econ and CS venues about choosing between two choices in numerous situations and from various perspectives.  While this setting is not all encompassing, it is a key special case and has been recognized as such over and over and over:  To name a few that cover diverse fields:
>
> -Dekel, Eddie, and Michele Piccione. "Sequential voting procedures in symmetric binary elections." Journal of political Economy, 2000.  (Social Choice).
>
> -Dasgupta, Anirban, and Arpita Ghosh. "Crowdsourced judgement elicitation with endogenous proficiency." WWW. 2013.  (Peer prediction)
>
> -Lobel, Ilan, and Evan Sadler. "Preferences, homophily, and social learning." Operations Research, 2016.
>
> In fact, some papers think it is so important, they do not even bother to explain it as a restriction: e.g.  Callander, Steven. "Bandwagons and momentum in sequential voting." The Review of Economic Studies 74.3 (2007): 653-684.
>
> These are all excellent papers and produced off the top of our heads, and we could find hundreds more.  The same “Results are somewhat narrow” critique could be applied to any of these.  The fact that all these apply only to making a choice between 2 alternatives is, indeed, a limitation of these papers (and ours).  However, it does not seem (to us) to follow that all their “results are somewhat narrow”.
>
> Moreover, this same critique of “somewhat narrow” could be applied to the works cited in our paper by Feddersen and Pesendorfer; Prelec, Seung, and McCoy; and the Marquis of Condorcet’s Jury Theorem.
>
>
> “Ex-ante/ex-post”;  “different from the more common goal of maximizing expected utility.” :
>
> Our choice of discussing ex-ante utilities in solution concept and ex-post utilities in the goal is motivated by practical applications. The underlying better alternative is unknown in the election phrase, so ex-ante utilities are considered for the solution concept. On the other hand, the ultimate goal is to select the alternative that *turns out to be better* after a certain period of time when the underlying better alternative is revealed (e.g., the performance of a selected president or the effect of a new policy in the next few years after the election), and in this case the ex-post utilities are considered.
>
> One cannot hope to maximize utility in a social choice setting such as this without strong additional assumptions.  The issue is that there might be 51% of agents that weakly prefer A to B (say utility epsilon versus utility 0) and 49% of agents that strongly prefer B to A  (say utility 0 versus utility 100).  In any anonymous mechanism, the former agents can lie to make sure A is chosen.  However, the social welfare lost by this decision could be unbounded (~100/epsilon). Therefore, optimal social welfare is not an achievable goal.
>
> It is interesting as you point out, that one way to interpret our results is that they are ex-post Bayesian efficient, as any strongly-truthful mechanism must yield to the wish of the majority.
>
>
> "In particular, this mechanism doesn’t seem appealing in the finite T case to agents who have, e.g., a weak preference to avoid false negative errors but a strong preference to avoid false positives.":
>
>  A key question here is: what is the alternative?  Basically, the majority vote.  (You can also use range voting, but that runs into huge incentive issues.)   You can also set another threshold like ⅔’s majority when you want to favor one outcome)  Does the majority vote work well in these cases?  In equilibrium, as Feddersen and Pesendorfer show, it should do nearly the same thing, just in a more convoluted way.  Agent’s will try to strategically manipulate their votes to take into account the beliefs and preferences of others.
>
>
> “About the revelation principle”:  Perhaps this is a semantic argument about what the “revelation principal” actually is.  At the end of the day, our mechanism implements a similar equilibrium as Feddersen and Pesendorfer’s Work while eliciting more information.
>
> However, merely invoking the revelation principle on Feddersen and Pesendorfer’s Work leaves one short of the achievements of this paper and requires new elements:
>
> 1. You note that *if* one can elicit enough information (sufficient statistics) then one can apply the revelation principle.  But what are sufficient statistics and how can they be elicited?  We show that they can be elicited in a very nice and natural way.  This does not follow from the mere invocation of the revelation principle, and requires the ideas of “surprisingly popular” and the “median trick” that are absent in Feddersen and Pesendorfer’s Work.
> 2. Our techniques enable us to obtain the much stronger guarantee of *strong* Bayes Nash Equilibrium, which does not follow from invoking the revelation principle on Feddersen and Pesendorfer’s Work .
> 3. The difference in the signal structures (discrete in our work vs. continuous in their work) make the nature of the two problems quite (technically) different.  So one cannot get from one to the other by a generic application of the principle.
>
>
> “Novelty of Theorem 3.1”: Theorem 3.1 is the result due to Prelec et al., we just include it for completeness (we have mentioned that the whole Sect. 3.1 is a literature review of Prelec et al.’s technique).  Perhaps you meant Theorem 3.2. Admittedly, it is not hard to show that our mechanism outputs the correct alternative when agents’ are truth-telling. Prelec et al.’s method can already guarantee this. However, our main focus here is to make the mechanism truthful (notice that Prelec et al’s method does not consider strategically behavored agents). Thus, we view Theorem 3.3 (or Theorem 3.4) as the main technical contribution of our paper.
>
> We should note that the impact of this paper is not predicted solely (or even mostly) on the technical difficulty of this work.  As explained in our response to reviewer 3:
>
>     “We see the simplicity of our work (and any work really) as a HUGE PLUS.   The simplicity means it should be implementable. This could be huge.  We cannot recall the last time someone suggested a voting scheme for two alternatives that is fundamentally different from the majority vote.  Most of the well-known mechanisms that have a big impact are those that are simple enough to appear in undergraduate textbooks.  We believe this is the type of big impact paper that NeurIPS is looking for.
>
>     “Moreover, the design of our mechanism may be simple, but it is not straightforward. Indeed, to the best of our knowledge, it is the first application of “surprisingly popular” to social choice in the 16 years since Bayesian Truth Serum (BTS) came out (Prelec, Seung, and McCoy’s surprisingly popular method can be seen as a simple application of BTS, which originally defined “surprisingly popular”).”
>
>
> Others: We also appreciate many of your other suggestions. Indeed, a lot of contents have to be deferred to the appendix due to the space limit. We will consider your suggestions carefully to adjust the arrangement of those contents.
>
>
> We hope that our response will enable you to raise your score.

---

> > ### Author Response · Authors · 2021-09-03
> > **Response to "Post Author Response"**
> >
> > We noticed that there is an update in your review with an extra "post author response" section. We appreciate that you read our response and share with us what you think. Here, we would like to further clarify a few things you mentioned.
> >
> > > Two alternatives - there is certainly plenty of work that looks at settings with two outcomes, but I still do think it is fair to call this restriction a limitation of the results in the paper.
> >
> > We agree it is a limitation, however we disagree that it makes the results narrow.
> >
> > > Ex-ante/Ex-post and Finite T - It seems like there are some worthwhile things to say here, and it would be good for some of them to make it into the paper.
> >
> > We will add some discussion of this to the final version.   Note that this also is what prior work, e.g., Feddersen and Pesendorfer, are doing: the equilibrium analysis is ex ante, but the goal—correct selection—is ex post.
> >
> > > imposing the signal structure
> >
> > Any partial information mechanism must describe the partial information structures for which it will work.  Our description is very standard in the peer-prediction literature, and closely related to the prior work (Feddersen and Pesendorfer) in the social choice literature.  Some restriction is necessary, for example, signals must carry some information.
> >
> > We would also like to remark that our mechanism is “detail-free”: the mechanism does not need to know the prior information $P_L$, $P_H$ and $P_{lL}, P_{lH}, P_{hL}, P_{hH}$.
> >
> > >It would strengthen the paper if theoretical results along these lines could be established.
> >
> > We are not sure that we entirely understand the type of statement that would be proven.  As you point out the “majority wish” is only defined ex post.  So we would really need something different, like social welfare, to base such a theory on.  However, no mechanism will do well versus social welfare in all settings.  For this reason the social choice rarely deals with social welfare (the distortion literature would be an exception, but this requires a new set of assumptions about embeddings).
> >
> > >Theorem 3.1
> >
> > We tried to make it clear that it is not our proof via the title of the section 3.1  and the preceding sentences “We will first review Prelec et al.’s Surprisingly Popular algorithm [Prelec et al., 2017], which works under a setting similar to ours but with non-strategic agents. Some part of the intuition behind our mechanism is based on Prelec et al.’s work.”  We can try to make it more clear.  In particular, the sentence which says  “The proof is deferred to Appendix C.2, but the intuition is straightforward.” is not the best.   This was added when we had to remove the full proof for length reasons.  However, it would have been better to say “For completeness, the full proof is appears in Appendix C.2, but the intuition is straightforward.”  While this particular sentence would not appear in a proceedings version (that do not have appendices) something analogous will.
> >
> > This also explains why we include the proof (sketch) of Theorem 3.1: the idea of Theorem 3.2 is based on Theorem 3.1, and we believe fully understanding the surprisingly popular method is helpful for understanding our mechanism.

---

### Decision · Program_Chairs · 2021-09-27

**Decision:**

Accept (Poster)

**Comment:**

There is not much substantive disagreement among the reviewers; the disagreement is mostly about how to value these results, in particular given the restriction to two alternatives.  (Such a restriction is common in social choice due to challenges that start to appear with three or more alternatives.)  The paper is left in a borderline position, but I'm inclined to give it the benefit of the doubt and recommend it for poster (in particular I think social choice results with two alternatives can be interesting and valuable).  The authors should take seriously though some of the reviewers' comments to improve the presentation of the paper.